# Comparisons of fatigue between dialysis modalities: A cross-sectional study

Yukio Maruyama[1]*, Masaaki Nakayama[2], Atsushi Ueda[3], Mariko Miyazaki[4], Takashi Yokoo[1]

1 Division of Nephrology and Hypertension, Department of Internal Medicine, The Jikei University School of Medicine, Tokyo, Japan, 2 Kidney Center, St Luke's International Hospital, Tokyo, Japan, 3 Department of Nephrology, Hitachi General Hospital, Ibaraki, Japan, 4 Research Division of Chronic Kidney Disease and Dialysis Treatment, Tohoku University Hospital, Miyagi, Japan

* maruyama@td5.so-net.ne.jp

**Data Availability Statement:** All relevant data are within the manuscript and its Supporting Information files.

**Funding:** This study was supported in part by research grants M.N. has received from Baxter

## Abstract

### Background

Fatigue is one of the most frequent complications in dialyzed patients and is associated with poorer patient outcomes. Multiple factors are reported to be associated with fatigue development. Of them, the impacts of dialysis modalities remain unknown.

### Methods

A total of 194 dialysis patients (mean age, 61±11 years; 134 males; modalities included hemodialysis (HD) in 26, online hemodiafiltration (HDF) in 74, peritoneal dialysis (PD) in 68, and combined therapy with PD and HD in 26 cases) were recruited for this cross-sectional study. Fatigue was assessed using the Profile of Mood States (POMS), a Visual Analogue Scale (VAS), and our original scale of fatigue, and depression was assessed by the Beck Depression Inventory-second edition (BDI-II). Our original scale of fatigue was administered both on dialysis and dialysis-free days to patients on HD and online HDF.

### Results

The scores of the POMS, VAS, and our original scale were weakly but significantly inter-related (rho = 0.58, P<0.01; rho = 0.47, P<0.01, and rho = 0.42, P<0.01 between POMS and VAS, POMS and our original scale for fatigue, and VAS and our original scale for fatigue, respectively). The scores of these 3 tests showed no significant differences among the 4 modalities. On multivariate analysis, age, body mass index, creatinine, and employment status were associated with the presence or severity of fatigue, whereas dialysis modality was not. A similar result was obtained in 122 patients without depression. The prevalence of fatigue by our original scale was significantly lower on dialysis-free days than on dialysis days in patients on HD and online HDF.

### Conclusions

The results suggest that there is no significant association between different dialysis modalities including HD, online HDF, PD and combined therapy with PD and HD and the prevalence or severity of fatigue.

International, Inc. (number 16CECPDAP0002). The funders had no role in study design, data collection and analysis, decision to publish, or preparation of the manuscript. There was no additional external funding received for this study.

**Competing interests:** I have read the journal's policy and the authors of this manuscript have the following competing interests: Y.M. and M.N. have received scholarship funds from Baxter International, Inc. and Terumo Corporation. No other authors have any conflicts of interest to declare. This does not alter our adherence to PLOS ONE policies on sharing data and materials.

## Introduction

Fatigue is one of the most frequent complications in patients receiving renal replacement therapy, and its prevalence has been reported to range from 42% to 89%, depending on patients' characteristics and assessment tools [1]. Notably, the presence of fatigue was associated with higher mortality in dialyzed patients [2–4]. Several factors, including physiological, psychological or behavioral, sociodemographic, and dialysis-related factors are associated with the development of fatigue [5]. Among them, physiological factors such as anemia, inflammation, malnutrition, uremia, and comorbidities are thought to be important because they are all modifiable. Depression, the most important psychological factor, is strongly associated with fatigue [6, 7] and is also associated with worse outcomes in patients on dialysis [8–10]. There have been several therapeutic strategies including both pharmacological and non-pharmacological approaches [1, 5].

Hemodialysis (HD) per se could induce fatigue through several factors such as bio-incompatibility of the dialyzer and dialysis apparatus which could induce production of pro-inflammatory cytokines or acute reduction of body fluid along with a decrease in blood pressure, whereas peritoneal dialysis (PD) per se could induced fatigue through dwelling of a bio-incompatible PD solution, chronic overhydration status, or a physiological burden [1, 5]. However, there have been only limited studies comparing the severity of fatigue between HD and PD patients [2, 11, 12]. Additionally, studies assessing the fatigue of patients receiving hemodiafiltration (HDF) have been very sparse [13].

The aim of this cross-sectional study was to compare the prevalence and severity of fatigue among patients receiving different dialysis modalities, including HD, online HDF, PD, and combined therapy with PD and HD. Furthermore, the state of depression was assessed in order to exclude possible psychological factors related to fatigue in these patients.

## Materials and methods

### Subjects

In this cross-sectional study, 194 patients receiving HD (n = 26, 13.4%), online HDF (n = 74, 38.1%), PD (n = 68, 35.1%), and combined therapy with PD and HD (n = 26, 13.4%) at Jikei University Hospital (Tokyo, Japan), the Jikei University Kashiwa Hospital (Chiba, Japan), the Jikei University Katsushika Medical Center (Tokyo, Japan), Shinagawa Kidney Clinic (Tokyo, Japan), Iidabashi Murai Clinic (Tokyo, Japan), Aoto Kidney Clinic (Tokyo, Japan), and Shin Kashiwa Clinic (Kashiwa, Chiba) were recruited between April 2018 and December 2018. Patients under 18 years of age, in addition to patients with a mental disorder, such as schizophrenia, depression or dementia, and those with terminal cancer were excluded. The ethics committees at Tohoku University Hospital and Jikei University Hospital approved this study protocol, and written, informed consent was obtained from all patients.

### Samples

Blood samples were collected once a month at regular outpatient visits from patients undergoing PD and at the start of HD treatment after the longest interval from those undergoing HD, online HDF, and combined therapy with PD and HD. Laboratory tests were conducted by standard laboratory techniques. Of note, approximately 20% of all patients had C-reactive protein (CRP) levels below the detectable levels, and the results of these patients are reported as being at the detectable levels. Each patient's employment status was determined using a self-reported questionnaire.

## Assessments of fatigue and depression

The patients' fatigue was evaluated using the Profile of Mood States (POMS), a Visual Analogue Scale (VAS), and our original scale of fatigue, whereas depression was assessed using the Beck Depression Inventory-second edition (BDI-II). All questionnaires were written format. The POMS scale is a questionnaire measuring six mood states: tension-anxiety (tension), depression-dejection (depression), anger-hostility (anger), fatigue-inertia (fatigue), vigor-activity (vigor), and confusion-bewilderment (confusion) on a five-point scale from 0 (not at all) to 4 (extreme); the fatigue subscale was used for the analysis. The VAS is a unidimensional scale in which a 100-mm line is anchored at either end by 'no tiredness at all' at the left end and 'complete exhaustion' at the other. The intensity of fatigue is measured in millimeters from the low (left) end of the scale. Our original scale of fatigue is a five-grade evaluation including Grade 1 (inexhaustible, feels well, and extremely active), Grade 2 (tireless, acts in the ordinary way without a sense of fatigue), Grade 3 (mild fatigue, acts in the ordinary way but feels tired), Grade 4 (moderate fatigue, feels tired with light work), and Grade 5 (intense fatigue, very tired, and falls asleep). Grades 3, 4 and 5 are regarded as fatigued states. We have already used this scale for the assessment of fatigue in two previous studies [14, 15]. Although the effectiveness of this scale has not yet been validated, we used this scale because of its usability and lower cost than other scales. Our original scale of fatigue was administered both on dialysis and dialysis-free days to patients on HD and online HDF.

The BDI–II is a 21-item, self-reported inventory designed to assess the presence and severity of depressive symptoms. Each item is rated on a 4-point scale ranging from 0 to 3, and the total score ranges from 0 to 63. A cutoff score of 14 indicates at least a mild-to-moderate level of depression [16, 17]. For all scales, higher scores indicate an increased level of fatigue or depression. Patients receiving PD or combined therapy with PD and HD answered all self-reported questionnaires at regular outpatient visits. Patients receiving HD or online HDF answered all self-reported questionnaires shortly after dialysis sessions.

## Statistical analyses

Data are presented as means ± standard deviation (SD) or medians and interquartile range (IQR), as appropriate. A p-value of less than 0.05 was considered significant. Differences among the four groups of renal dialysis were analyzed using one-way analysis of variance (ANOVA) or the non-parametric Kruskal-Wallis test, as appropriate. Differences were considered significant when the F-value was less than 0.05. The associations between each clinical parameter were calculated using Spearman's correlation coefficient (rho). Independent factors affecting fatigue were assessed using a logistic regression model. In these analyses, the objective variables were dichotomous variables, including POMS score $\geq$ 10, VAS score $\geq$ 48, and our original scale of fatigue score $\geq$ 3. The changes in our original scale of fatigue between dialysis and dialysis-free day were evaluated using a paired t-test. The cut-off values of the POMS and VAS scores were based on receiver operating characteristic analysis, in which fatigue was defined as our original scale of fatigue score $\geq$ 3. On multivariate analysis, several confounding factors reported to be associated with fatigue in dialyzed patients, including age, sex, dialysis duration, state of employment, the presence of cardiovascular disease and diabetes, body mass index (BMI), hemoglobin (Hb), and serum albumin, were included. Additionally, confounding factor that had shown an appreciable association (P<0.1) with a measure of fatigue were also included in the multivariate analysis. Of them, dialysis duration was markedly skewed and was therefore log-transformed to normalize its distribution. Additionally, independent factors affecting fatigue were assessed using a multiple regression model with the same confounding factors. In these analyses, POMS, VAS, and our original scale of fatigue were treated as

continuous variables, and the POMS score was log-transformed to normalize its distribution. Sensitivity analyses including 122 patients without depression (BDI-II score < 14) were performed. Data were statistically analyzed using STATA version 16.0 (STATA Corporation, College Station, TX, USA).

## Results

Table 1 shows the patients' demographic and biochemical characteristics. The age of the patients was 61 ± 11 years, and males predominated (69.1%). Diabetic patients accounted for 27.3%. Age, sex, prevalence of diabetes, and BMI were not significantly different among the four groups. The duration of dialysis was longer in the HD and online HDF groups (129 (42–259), 153 (96–228), 25 (9–41), and 58 (35–74) months for HD, online HDF, PD, and combined therapy, respectively). Hb was higher in patients receiving combined therapy. Serum creatinine was higher in patients receiving online HDF and combined therapy. Urea nitrogen was higher in patients receiving online HDF. Serum albumin was lower in patients receiving PD

**Table 1. Patients' characteristics.**

|  | All patients | HD | Online HDF | PD | Combined Tx | P |
|---|---|---|---|---|---|---|
| Number (%) | 194 | 26 (13.4%) | 74 (38.1%) | 68 (35.1%) | 26 (13.4%) |  |
| Age (y) | 61±11 | 66±9 | 59±10 | 60±12 | 62±11 | 0.06 |
| Male sex (%) | 134 (69.1%) | 19 (73.1%) | 46 (62.2%) | 51 (75.0%) | 18 (69.2%) | 0.40 |
| Duration of dialysis (months) | 64 (27–152) | 129 (42–259) | 153 (96–228) | 25 (9–41) | 58 (35–74) | <0.01 |
| Employed (%) | 104 (53.9%) | 9 (34.6%) | 45 (60.8%) | 38 (55.9%) | 12 (48.0%) | 0.12 |
| Smoker (%) | 17 (8.9%) | 0 (0%) | 11 (14.9%) | 6 (8.8%) | 0 (0%) | 0.04 |
| History of CVD (%) | 57 (29.4%) | 12 (46.2%) | 17 (23.0%) | 19 (27.9%) | 9 (34.6%) | 0.14 |
| Diabetes (%) | 53 (27.3%) | 10 (38.5%) | 18 (24.3%) | 17 (25.0%) | 8 (30.8%) | 0.51 |
| Height (cm) | 165±9 | 164±9 | 165±9 | 165±9 | 164±10 | 0.86 |
| BW (kg) | 62.8±14.4 | 61.9±17.9 | 61.6±14.8 | 64.3±13.1 | 63.0±13.3 | 0.73 |
| BMI (kg/m$^2$) | 23.0±4.4 | 22.8±5.3 | 22.4±4.5 | 23.6±4.2 | 23.3±3.8 | 0.40 |
| Systolic BP (mmHg) | 142±17 | 140±16 | 149±19 | 136±14 | 141±16 | <0.01 |
| Diastolic BP (mmHg) | 80±12 | 72±13 | 82±12 | 80±11 | 84±11 | <0.01 |
| Laboratory findings |  |  |  |  |  |  |
| Hemoglobin (g/dL) | 11.3±1.2 | 11.2±1.0 | 11.1±1.0 | 11.4±1.2 | 11.8±1.6 | 0.04 |
| Creatinine (mg/dL) | 11.5±2.8 | 10.9±2.3 | 12.4±2.3 | 10.5±2.9 | 12.3±3.2 | <0.01 |
| Urea nitrogen (mg/dL) | 59.7±12.5 | 59.7±12.3 | 64.9±12.0 | 55.1±13.5 | 56.7±14.4 | <0.01 |
| Serum albumin (g/dL) | 3.5±0.4 | 3.8±0.4 | 3.7±0.3 | 3.3±0.5 | 3.1±0.3 | <0.01 |
| CRP (mg/dL) | 0.10 (0.05–0.27) | 0.17 (0.04–0.26) | 0.09 (0.01–0.22) | 0.10 (0.05–0.43) | 0.10 (0.10–0.29) | 0.41 |
| β2 microglobulin (mg/L) | 27.3±7.4 | 28.7±5.8 | 28.2±4.1 | 25.7±0.9 | 29.7±1.5 | 0.09 |
| Use of ESAs (%) | 171 (88.1%) | 21 (80.8%) | 61 (82.4%) | 64 (94.1%) | 25 (96.2%) | 0.06 |
| POMS | 11.3±4.1 | 11.6±3.9 | 11.2±3.8 | 11.1±4.6 | 12.0±3.5 | 0.76 |
| VAS | 48 (27–63) | 49 (39–56) | 46 (21–62) | 48 (27–63) | 51 (41–63) | 0.43 |
| Our original scale of fatigue | 3.0±1.0 | 3.3±0.9 | 2.9±1.1 | 2.9±0.9 | 3.5±0.9 | 0.02 |
| Our original scale of fatigue (dialysis-free day) | 2.4±0.8 | 2.6±0.8 | 2.3±0.7 | N.A. | N.A. | 0.08 |
| BDI-II | 11 (7–17) | 11 (10–13) | 10 (6–16) | 13 (6–18) | 11 (9–19) | 0.33 |

* n = 168, for values below the detection level, the detection level was used.

Abbreviations: HD, hemodialysis; HDF, hemodiafiltration; PD, peritoneal dialysis; Tx, therapy; CVD, cardiovascular disease; BW, body weight; BMI, body mass index; BP, blood pressure; CRP, C-reactive protein; ESAs, erythropoiesis-stimulating agents; N.A., not applicable; POMS, Profile of Mood States; VAS, Visual Analogue Scale; BDI-II, the Beck Depression Inventory-second edition.

POMS (P=0.46)

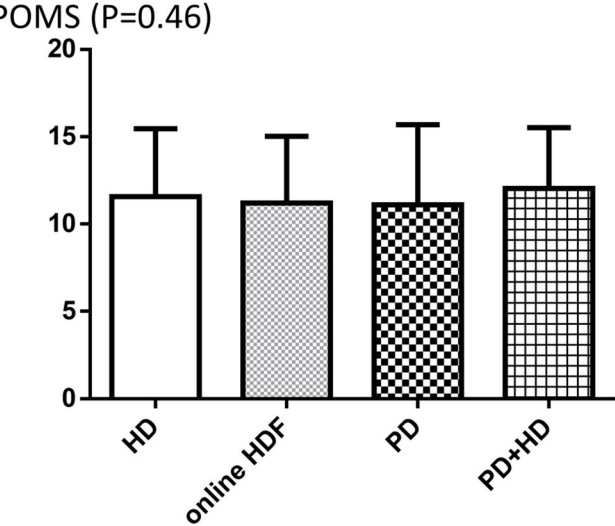

VAS (P=0.43)

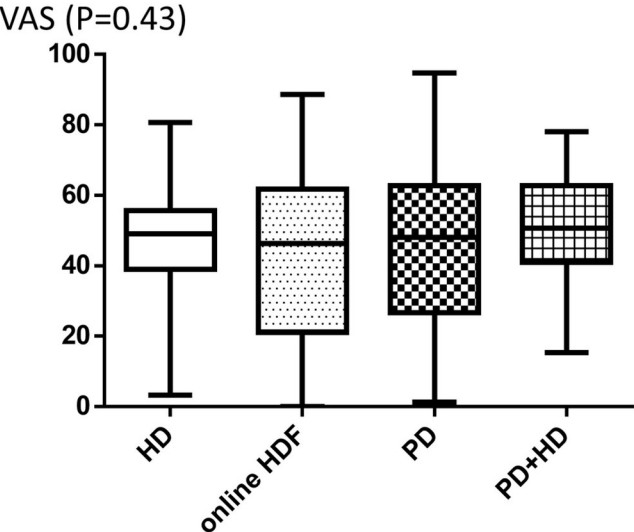

Our original scale of fatigue (P=0.01)

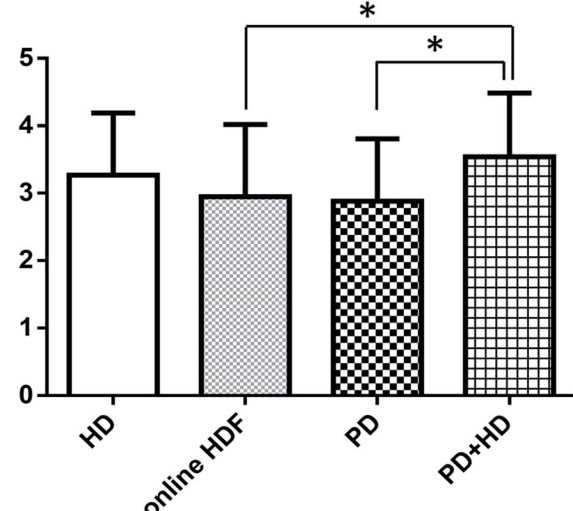

**Fig 1. Comparisons of the scores of the POMS, VAS, and our original scale of fatigue in patients receiving HD, online HDF, PD, and combined therapy with PD and HD.** Abbreviations: POMS, Profile of Mood States; VAS, Visual Analogue Scale; HD, hemodialysis; HDF, hemodiafiltration; PD, peritoneal dialysis. * indicates significance.

and combined therapy. CRP and β2 microglobulin levels showed no significant differences among the four groups.

All participants read and responded to the questionnaires themselves. According to the results of the self-reported questionnaire, 104 (53.9%) patients were employed. The percentage of employment was not significantly different among the four groups. Of the 104 patients at work, 92 (88.5%) patients felt that the fatigue affected their work to mild (n = 48, 47.1%), moderate (n = 28, 26.9%), and severe (n = 16, 15.4%) degrees, whereas 10 (9.6%) patients felt that the fatigue did not affect their work. Of the 90 unemployed patients, 75 (83.3%) patients felt that the fatigue affected their daily life to mild (n = 28, 31.1%), moderate (n = 32, 35.6%), and severe (n = 15, 16.7%) degrees, whereas 10 (11.1%) patients felt that the fatigue did not affect their daily life. However, 22 (24.4%) patients wished to work.

The scores of the three assessments for fatigue including POMS, VAS, and our original scale of fatigue were significantly inter-related (rho = 0.58, P<0.01; rho = 0.47, P<0.01, and rho = 0.42, P<0.01 between POMS and VAS, POMS and our original scale for fatigue, and VAS and our original scale for fatigue, respectively). The scores on the POMS and VAS were not significantly different among the four groups, whereas those of our original scale of fatigue were higher in patients receiving combined therapy with PD and HD (Table 1, Fig 1). In the analysis using our original scale of fatigue, the prevalence of fatigue, defined as Grades 3, 4 and 5, was significantly lower on dialysis-free days than on dialysis days in patients on HD and online HDF (50.0% vs. 84.6%; P<0.01 and 32.4% vs. 66.2%; P<0.01 for patients on HD and online HDF, respectively). Similarly, the severity of fatigue, indicated by the scores of this scale, was significantly lower on dialysis-free days than on dialysis days (2.6±0.8 vs. 3.3±0.9; P<0.01 and 2.3±0.7 vs. 2.9±1.1; P<0.01 for patients on HD and online HDF, respectively).

Table 2 shows multiple logistic regression models for the results of the three tests, and the modality of dialysis did not affect the presence of fatigue. Instead, employed patients had a significantly lower risk of the presence of fatigue on the VAS and our original scale of fatigue. In addition, BMI, age and the use of erythropoiesis-stimulating agents were associated with the presence of fatigue on the POMS, VAS score and our original scale for fatigue, respectively. The results of the multiple regression model, with analysis using the findings of the three tests as continuous variables, were very similar (Table 3). The modality of dialysis did not affect the severity of fatigue. Instead, employed patients had a significantly lower severity of fatigue on the VAS and our original scale of fatigue. BMI was associated with the severity of fatigue on the POMS score, whereas serum creatinine was associated with the severity of fatigue on the POMS and VAS scores.

Sensitivity analyses including 122 patients without depression (BDI-II score < 14) were performed. The results of the logistic regression models (Table 4) and those of the multiple regression models (Table 5) were very similar.

## Discussion

In this cross-sectional study, the prevalence and the severity of fatigue were not significantly different among patients receiving different dialysis modalities (HD, online HDF, PD, and combined therapy with PD and HD). There was a particular focus on the effect of the presence of depression in the present study because fatigue and depression are closely interrelated, and depression may manifest as feelings of tiredness and lack of energy [1, 5]. The fatigue profile

**Table 2. Logistic regression model for higher scores for fatigue.**

| | | Unadjusted | | Adjusted | |
|---|---|---|---|---|---|
| | | Odds ratio (95% CI) | P value | Odds ratio (95% CI) | P value |
| POMS | | | | | |
| | Age | 1.001 (0.975 to 1.028) | 0.92 | 0.981 (0.950 to 1.014) | 0.25 |
| | Male gender | 0.551 (0.285 to 1.063) | 0.08 | 0.729 (0.336 to 1.579) | 0.42 |
| | Dialysis duration * | 1.106 (0.872 to 1.403) | 0.41 | 1.032 (0.724 to 1.471) | 0.86 |
| | Employed | 0.602 (0.333 to 1.088) | 0.09 | 0.668 (0.321 to 1.391) | 0.28 |
| | Smoker | 0.655 (0.241 to 1.781) | 0.41 | | |
| | Dialysis modality | | | | |
| | HD | Ref. | | Ref. | |
| | Online HDF | 0.870 (0.342 to 2.215) | 0.77 | 0.760 (0.267 to 2.164) | 0.61 |
| | PD | 0.712 (0.278 to 1.823) | 0.48 | 1.013 (0.295 to 3.478) | 0.98 |
| | Combined therapy | 1.437 (0.439 to 4.699) | 0.55 | 1.913 (0.445 to 8.218) | 0.38 |
| | CVD | 0.766 (0.407 to 1.439) | 0.41 | 0.591 (0.288 to 1.216) | 0.15 |
| | Diabetes | 1.244 (0.642 to 2.410) | 0.52 | 1.976 (0.908 to 4.296) | 0.09 |
| | BMI | 0.922 (0.862 to 0.986) | 0.02 | 0.895 (0.825 to 0.970) | 0.01 |
| | Hemoglobin | 0.901 (0.704 to 1.153) | 0.41 | 0.820 (0.619 to 1.087) | 0.17 |
| | Creatinine | 1.037 (0.955 to 1.126) | 0.39 | | |
| | Urea nitrogen | 0.996 (0.975 to 1.018) | 0.72 | | |
| | Serum albumin | 1.057 (0.549 to 2.033) | 0.87 | 1.425 (0.566 to 3.587) | 0.45 |
| | CRP * | 0.979 (0.755 to 1.268) | 0.87 | | |
| | β2 microglobulin | 1.004 (0.959 to 1.052) | 0.87 | | |
| | Use of ESAs | 1.319 (0.546 to 3.182) | 0.54 | | |
| VAS | | | | | |
| | Age | 0.992 (0.967 to 1.018) | 0.56 | 0.966 (0.934 to 0.998) | 0.04 |
| | Male gender | 0.910 (0.492 to 1.682) | 0.76 | 1.578 (0.744 to 3.351) | 0.24 |
| | Dialysis duration * | 1.173 (0.928 to 1.483) | 0.18 | 1.384 (0.964 to 1.986) | 0.08 |
| | Employed | 0.466 (0.260 to 0.832) | 0.01 | 0.370 (0.176 to 0.776) | <0.01 |
| | Smoker | 1.186 (0.423 to 3.327) | 0.75 | | |
| | Dialysis modality | | | | |
| | HD | Ref. | | Ref. | |
| | Online HDF | 0.769 (0.314 to 1.884) | 0.57 | 0.629 (0.229 to 1.728) | 0.37 |
| | PD | 0.938 (0.378 to 2.324) | 0.89 | 1.166 (0.342 to 3.978) | 0.81 |
| | Combined therapy | 1.821 (0.582 to 5.698) | 0.30 | 1.651 (0.399 to 6.837) | 0.49 |
| | CVD | 0.820 (0.441 to 1.524) | 0.53 | 0.636 (0.312 to 1.295) | 0.21 |
| | Diabetes | 0.963 (0.509 to 1.822) | 0.91 | 1.060 (0.498 to 2.256) | 0.88 |
| | BMI | 0.964 (0.903 to 1.028) | 0.27 | 0.959 (0.887 to 1.038) | 0.30 |
| | Hemoglobin | 1.129 (0.887 to 1.436) | 0.33 | 1.065 (0.808 to 1.404) | 0.65 |
| | Creatinine | 1.039 (0.962 to 1.122) | 0.33 | | |
| | Urea nitrogen | 0.986 (0.965 to 1.007) | 0.20 | | |
| | Serum albumin | 0.612 (0.319 to 1.175) | 0.14 | 0.713 (0.285 to 1.787) | 0.47 |
| | CRP * | 1.192 (0.919 to 1.546) | 0.19 | | |
| | β2 microglobulin | 1.041 (0.993 to 1.090) | 0.10 | | |
| | Use of ESAs | 0.682 (0.28 to 1.661) | 0.40 | | |
| Our original scale of fatigue | | | | | |
| | Age | 1.014 (0.986 to 1.043) | 0.34 | 0.984 (0.95 to 1.020) | 0.38 |
| | Male gender | 0.855 (0.432 to 1.689) | 0.65 | 1.258 (0.544 to 2.907) | 0.59 |
| | Dialysis duration * | 1.138 (0.884 to 1.464) | 0.32 | 1.136 (0.770 to 1.677) | 0.52 |

(*Continued*)

**Table 2.** (Continued)

| | | Unadjusted | | Adjusted | |
|---|---|---|---|---|---|
| | | Odds ratio (95% CI) | P value | Odds ratio (95% CI) | P value |
| | Employed | 0.394 (0.203 to 0.762) | 0.01 | 0.433 (0.190 to 0.987) | 0.047 |
| | Smoker | 0.319 (0.116 to 0.877) | 0.03 | 0.373 (0.113 to 1.226) | 0.10 |
| | Dialysis modality | | | | |
| | HD | Ref. | | Ref. | |
| | Online HDF | 0.356 (0.111 to 1.147) | 0.08 | 0.422 (0.113 to 1.573) | 0.20 |
| | PD | 0.333 (0.103 to 1.080) | 0.07 | 0.649 (0.149 to 2.820) | 0.56 |
| | Combined therapy | 1.394 (0.279 to 6.953) | 0.69 | 1.994 (0.313 to 12.701) | 0.47 |
| | CVD | 0.936 (0.475 to 1.845) | 0.85 | 0.650 (0.289 to 1.464) | 0.30 |
| | Diabetes | 1.550 (0.743 to 3.233) | 0.24 | 1.625 (0.673 to 3.924) | 0.28 |
| | BMI | 0.974 (0.908 to 1.043) | 0.45 | 0.940 (0.861 to 1.026) | 0.17 |
| | Hemoglobin | 1.187 (0.910 to 1.549) | 0.21 | 0.973 (0.699 to 1.352) | 0.87 |
| | Creatinine | 0.976 (0.922 to 1.033) | 0.40 | | |
| | Urea nitrogen | 1.006 (0.983 to 1.029) | 0.61 | | |
| | Serum albumin | 0.942 (0.467 to 1.901) | 0.87 | 1.265 (0.452 to 3.535) | 0.65 |
| | CRP* | 1.183 (0.886 to 1.580) | 0.26 | | |
| | β2 microglobulin | 1.032 (0.982 to 1.085) | 0.21 | | |
| | Use of ESAs | 0.334 (0.095 to 1.173) | 0.09 | 0.222 (0.052 to 0.951) | 0.04 |

Abbreviations: CI, confidence interval; POMS, Profile of Mood States; HD, hemodialysis; HDF, hemodiafiltration; PD, peritoneal dialysis; CVD, cardiovascular disease; BMI, body mass index; CRP, C-reactive protein; ESAs, erythropoiesis-stimulating agents; VAS, Visual Analogue Scale.

* Dialysis duration and CRP are log-transformed.

was examined in 122 patients without depression in a sensitivity analysis, and the prevalence and the severity of fatigue were not significantly different among patients receiving different dialysis modalities. Additionally, the prevalence of fatigue by our original scale was significantly lower on dialysis-free days than on dialysis days in patients on HD and online HDF.

There have been several studies comparing the severity of fatigue between HD and PD patients. Zazzeroni et al. [11] conducted a systematic review and meta-analysis including 7 studies and 1857 dialyzed patients, of whom 1165 were undergoing HD and 692 were undergoing PD, and they found that the scores of the energy/fatigue subscales were not different between the two groups. Jhamb et al. [2] reported that the SF-36 vitality scales at baseline and 1-year after dialysis initiation were not significantly different between incident HD and PD patients. In the present study, there were no significant differences in the fatigue scales between HD and PD, and the results were consistent with those prior reports.

To the best of our knowledge, this was the first study to examine fatigue in patients receiving online HDF or combined therapy with PD and HD. Online HDF, where replacement fluid is prepared by further purifying dialysate fluid instead of manufacturer-provided solutions, has made HDF more practical and more cost-effective, and high-volume online HDF could potentially improve symptoms, reduce morbidity, and may even improve survival [18]. The number of patients treated with online HDF has been increasing after the 2012 revision to the medical reimbursement system, and it has reached 70000, approximately 30% of all dialysis patients at the end of 2017 in Japan [19]. Suwabe et al. [13] conducted a systematic review and meta-analysis to compare QOL between patients receiving online HDF and HD and concluded that online HDF was associated with a non-significant decrease in the fatigue score compared to HD. Of note, online HDF with post dilution was performed in all of the studies,

**Table 3. Multiple regression model for the scales of fatigue.**

| Variable | Unadjusted | | | | Adjusted | | | |
|---|---|---|---|---|---|---|---|---|
| | Regression coefficient | 95%CI | t value | p value | Regression coefficient | 95%CI | t value | p value |
| POMS (log-transformed) | | | | | | | | |
| Age | 0.0002 | -0.0043 to 0.0048 | 0.10 | 0.92 | -0.0015 | -0.0070 to 0.0039 | -0.56 | 0.57 |
| Male gender | -0.0639 | -0.1723 to 0.0445 | -1.16 | 0.25 | -0.0088 | -0.1322 to 0.1147 | -0.14 | 0.89 |
| Dialysis duration * | 0.0146 | -0.0264 to 0.0556 | 0.70 | 0.48 | -0.0111 | -0.0713 to 0.0491 | -0.36 | 0.72 |
| Employed | -0.0824 | -0.1831 to 0.0183 | -1.61 | 0.11 | -0.1020 | -0.2230 to 0.0191 | -1.66 | 0.10 |
| Smoker | -0.016 | -0.195 to 0.162 | -0.18 | 0.86 | | | | |
| Dialysis modality | | | | | | | | |
| HD | Ref. | | | | Ref. | | | |
| Online HDF | -0.0290 | -0.1884 to 0.1303 | -0.36 | 0.72 | -0.0493 | -0.2179 to 0.1192 | -0.58 | 0.56 |
| PD | -0.0637 | -0.2249 to 0.0975 | -0.78 | 0.44 | -0.0517 | -0.2546 to 0.1512 | -0.5 | 0.62 |
| Combined therapy | 0.0633 | -0.1306 to 0.2572 | 0.64 | 0.52 | 0.0757 | -0.1573 to 0.3086 | 0.64 | 0.52 |
| CVD | -0.0583 | -0.1683 to 0.0518 | -1.04 | 0.30 | -0.0834 | -0.2014 to 0.0347 | -1.39 | 0.17 |
| Diabetes | -0.0291 | -0.1418 to 0.0837 | -0.51 | 0.61 | 0.0186 | -0.1057 to 0.1429 | 0.29 | 0.77 |
| BMI | -0.0115 | -0.0229 to -0.0002 | -2.01 | 0.046 | -0.0149 | -0.0279 to -0.0019 | -2.26 | 0.03 |
| Hemoglobin | -0.0154 | -0.0578 to 0.0270 | -0.72 | 0.48 | -0.0240 | -0.0694 to 0.0213 | -1.05 | 0.30 |
| Creatinine | 0.00889 | -0.00057 to 0.01834 | 1.85 | 0.07 | 0.01117 | 0.00088 to 0.02146 | 2.14 | 0.03 |
| Urea nitrogen | -0.00022 | -0.00394 to 0.00351 | -0.12 | 0.91 | | | | |
| Serum albumin | -0.0137 | -0.1274 to 0.1001 | -0.24 | 0.81 | 0.0135 | -0.1367 to 0.1637 | 0.18 | 0.86 |
| CRP * | -0.0005 | -0.0469 to 0.0458 | -0.02 | 0.98 | | | | |
| β2 microglobulin | 0.00515 | -0.00323 to 0.01353 | 1.22 | 0.23 | | | | |
| Use of ESAs | 0.0724 | -0.0828 to 0.2275 | 0.92 | 0.36 | | | | |
| VAS | | | | | | | | |
| Age | -0.138 | -0.425 to 0.150 | -0.94 | 0.35 | -0.274 | -0.616 to 0.069 | -1.58 | 0.12 |
| Male gender | 0.00 | -6.84 to 6.84 | 0.00 | 1.00 | 3.32 | -4.36 to 11.00 | 0.85 | 0.40 |
| Dialysis duration * | 0.59 | -1.99 to 3.18 | 0.45 | 0.65 | 1.17 | -2.59 to 4.92 | 0.61 | 0.54 |
| Employed | -6.63 | -12.92 to -0.34 | -2.08 | 0.04 | -9.73 | -17.34 to -2.13 | -2.53 | 0.01 |
| Smoker | 3.63 | -7.74 to 15.00 | 0.63 | 0.53 | | | | |
| Dialysis modality | | | | | | | | |
| HD | Ref. | | | | Ref. | | | |
| Online HDF | -2.95 | -12.90 to 6.99 | -0.59 | 0.56 | -3.70 | -14.19 to 6.79 | -0.70 | 0.49 |
| PD | 0.19 | -9.88 to 10.27 | 0.04 | 0.97 | 0.95 | -11.74 to 13.64 | 0.15 | 0.88 |
| Combined therapy | 5.67 | -6.55 to 17.88 | 0.92 | 0.36 | 3.71 | -10.80 to 18.21 | 0.50 | 0.61 |
| CVD | -0.08 | -6.99 to 6.82 | -0.02 | 0.98 | -1.21 | -8.60 to 6.18 | -0.32 | 0.75 |
| Diabetes | 2.08 | -5.02 to 9.17 | 0.58 | 0.57 | 2.88 | -4.93 to 10.68 | 0.73 | 0.47 |
| BMI | -0.122 | -0.838 to 0.593 | -0.34 | 0.74 | -0.439 | -1.251 to 0.373 | -1.07 | 0.29 |
| Hemoglobin | 1.30 | -1.35 to 3.95 | 0.97 | 0.34 | 0.89 | -1.95 to 3.72 | 0.62 | 0.54 |
| Creatinine | 0.698 | 0.110 to 1.287 | 2.34 | 0.02 | 0.804 | 0.163 to 1.444 | 2.47 | 0.01 |
| Urea nitrogen | -0.188 | -0.423 to 0.047 | -1.58 | 0.12 | | | | |
| Serum albumin | -4.38 | -11.45 to 2.69 | -1.22 | 0.22 | -2.29 | -11.64 to 7.05 | -0.48 | 0.63 |
| CRP * | 1.99 | -0.81 to 4.80 | 1.40 | 0.16 | | | | |
| β2 microglobulin | 0.325 | -0.190 to 0.840 | 1.25 | 0.21 | | | | |
| Use of ESAs | -0.26 | -9.98 to 9.45 | -0.05 | 0.96 | | | | |
| Our original scale of fatigue | | | | | | | | |
| Age | 0.00995 | -0.00290 to 0.02281 | 1.53 | 0.13 | -0.00397 | -0.01915 to 0.01120 | -0.52 | 0.61 |
| Male gender | -0.126 | -0.434 to 0.182 | -0.81 | 0.42 | 0.115 | -0.230 to 0.460 | 0.66 | 0.51 |

*(Continued)*

**Table 3.** (Continued)

| Variable | Unadjusted | | | | Adjusted | | | |
|---|---|---|---|---|---|---|---|---|
| | Regression coefficient | 95%CI | t value | p value | Regression coefficient | 95%CI | t value | p value |
| Dialysis duration * | 0.0775 | -0.0389 to 0.1938 | 1.31 | 0.19 | 0.1007 | -0.0664 to 0.2678 | 1.19 | 0.24 |
| Employed | -0.465 | -0.744 to -0.186 | -3.29 | 0.00 | -0.339 | -0.679 to -0.0003 | -1.98 | 0.050 |
| Smoker | -0.433 | -0.932 to 0.065 | -1.72 | 0.09 | -0.230 | -0.756 to 0.296 | -0.86 | 0.39 |
| Dialysis modality | | | | | | | | |
| HD | Ref. | | | | Ref. | | | |
| Online HDF | -0.323 | -0.766 to 0.120 | -1.44 | 0.15 | -0.234 | -0.707 to 0.239 | -0.98 | 0.33 |
| PD | -0.387 | -0.835 to 0.061 | -1.70 | 0.09 | -0.142 | -0.708 to 0.424 | -0.50 | 0.62 |
| Combined therapy | 0.269 | -0.270 to 0.808 | 0.99 | 0.33 | 0.243 | -0.408 to 0.893 | 0.74 | 0.46 |
| CVD | -0.066 | -0.379 to 0.247 | -0.41 | 0.68 | -0.217 | -0.547 to 0.113 | -1.30 | 0.20 |
| Diabetes | 0.274 | -0.044 to 0.591 | 1.70 | 0.09 | 0.257 | -0.091 to 0.605 | 1.46 | 0.15 |
| BMI | -0.0126 | -0.0451 to 0.0198 | -0.77 | 0.44 | -0.0135 | -0.0496 to 0.0226 | -0.74 | 0.46 |
| Hemoglobin | 0.0468 | -0.0733 to 0.1669 | 0.77 | 0.44 | -0.0017 | -0.1290 to 0.1255 | -0.03 | 0.98 |
| Creatinine | -0.0254 | -0.0522 to 0.0014 | -1.87 | 0.06 | -0.0202 | -0.0488 to 0.0084 | -1.39 | 0.17 |
| Urea nitrogen | -0.00297 | -0.01353 to 0.00758 | -0.56 | 0.58 | | | | |
| Serum albumin | -0.140 | -0.459 to 0.179 | -0.87 | 0.39 | 0.005 | -0.414 to 0.425 | 0.03 | 0.98 |
| CRP * | 0.0403 | -0.0843 to 0.1648 | 0.64 | 0.52 | | | | |
| β2 microglobulin | 0.0103 | -0.0122 to 0.0329 | 0.91 | 0.37 | | | | |
| Use of ESAs | -0.145 | -0.585 to 0.296 | -0.65 | 0.52 | | | | |

Abbreviations: CI, confidence interval; POMS, Profile of Mood States; HD, hemodialysis; HDF, hemodiafiltration; PD, peritoneal dialysis; CVD, cardiovascular disease; BMI, body mass index; CRP, C-reactive protein; ESAs, erythropoiesis-stimulating agents; VAS, Visual Analogue Scale.

* Dialysis duration and CRP are log-transformed.

whereas more than 95% of patients used predilution online HDF in Japan because of the lower blood flow rate and lower serum albumin levels [19]. Because the predilution method is considered advantageous due to the availability of a large volume of substitution solution and lower albumin leakage than the post-dilution method, it could have a good effect on the prevention of fatigue. Although a PD first policy is recommended in order to maximize the advantages of PD therapy including excellent comparable survival, lower cost and improved QOL [20, 21], its long-term continuation is difficult because of a decrease in residual renal function and deterioration of the peritoneal membrane. Combined therapy with PD and HD was introduced in Japan in the 1990s and became popular as the preferred dialysis modality of Japan. The combined therapy, generally in the form of 5–6 days of PD and one HD session per week, is the treatment option for PD patients with inadequate dialysis and/or fluid overload instead of a complete shift to HD alone, and approximately 1900 patients (20% of all patients on PD) are receiving this therapy in Japan [22, 23]. This unique modality is expected to reduce fatigue compared to direct transfer to HD because of a decrease in the number of HD sessions. However, in the current study, there was no significant difference in the fatigue scales between patients receiving online HDF and combined therapy with PD and HD.

While dialysis modalities were not associated with fatigue in dialysis patients in the present study, several influential factors were identified by multivariate analysis, such as age, BMI, creatinine, and employment status. The present findings were consistent with a prior report that showed employment status to be strongly associated with fatigue and depression in HD patients [24–26]. In the present study, more than 80% of patients felt that the fatigue affected their work or daily life, including those who were unemployed, but wished to work. Thus, it is

**Table 4. Logistic regression model for higher scores for fatigue in patients without depression.**

| | | Unadjusted | | Adjusted | |
|---|---|---|---|---|---|
| | | Odds ratio (95% CI) | P value | Odds ratio (95% CI) | P value |
| POMS | | | | | |
| | Age | 0.998 (0.965 to 1.032) | 0.90 | 0.965 (0.920 to 1.011) | 0.14 |
| | Male gender | 0.581 (0.258 to 1.309) | 0.19 | 0.829 (0.308 to 2.229) | 0.71 |
| | Dialysis duration * | 1.076 (0.808 to 1.434) | 0.62 | 0.975 (0.648 to 1.468) | 0.91 |
| | Employed | 0.556 (0.263 to 1.172) | 0.12 | 0.499 (0.196 to 1.269) | 0.14 |
| | Smoker | 0.465 (0.111 to 1.953) | 0.30 | | |
| | Dialysis modality | | | | |
| | HD | Ref. | | Ref. | |
| | Online HDF | 0.818 (0.289 to 2.317) | 0.71 | 0.647 (0.192 to 2.181) | 0.48 |
| | PD | 0.662 (0.223 to 1.967) | 0.46 | 0.951 (0.228 to 3.966) | 0.95 |
| | Combined therapy | 1.473 (0.362 to 5.996) | 0.59 | 2.186 (0.343 to 13.926) | 0.41 |
| | CVD | 0.745 (0.334 to 1.660) | 0.47 | 0.597 (0.242 to 1.474) | 0.26 |
| | Diabetes | 1.227 (0.551 to 2.735) | 0.62 | 2.196 (0.821 to 5.871) | 0.12 |
| | BMI | 0.939 (0.865 to 1.020) | 0.14 | 0.895 (0.802 to 0.998) | 0.045 |
| | Hemoglobin | 0.839 (0.607 to 1.159) | 0.29 | 0.747 (0.512 to 1.088) | 0.13 |
| | Creatinine | 0.992 (0.873 to 1.127) | 0.90 | | |
| | Urea nitrogen | 0.989 (0.962 to 1.016) | 0.43 | | |
| | Serum albumin | 1.214 (0.503 to 2.931) | 0.67 | 1.316 (0.355 to 4.877) | 0.68 |
| | CRP * | 0.988 (0.693 to 1.409) | 0.95 | | |
| | β2 microglobulin | 1.014 (0.954 to 1.077) | 0.66 | | |
| | Use of ESAs | 1.209 (0.409 to 3.570) | 0.73 | | |
| VAS | | | | | |
| | Age | 0.993 (0.959 to 1.028) | 0.69 | 0.959 (0.911 to 1.009) | 0.11 |
| | Male gender | 0.933 (0.412 to 2.111) | 0.87 | 1.329 (0.473 to 3.734) | 0.59 |
| | Dialysis duration * | 1.290 (0.949 to 1.753) | 0.10 | 1.321 (0.852 to 2.048) | 0.21 |
| | Employed | 0.459 (0.215 to 0.979) | 0.04 | 0.372 (0.141 to 0.984) | 0.046 |
| | Smoker | 0.44 (0.085 to 2.277) | 0.33 | | |
| | Dialysis modality | | | | |
| | HD | Ref. | | Ref. | |
| | Online HDF | 0.501 (0.176 to 1.433) | 0.20 | 0.342 (0.099 to 1.182) | 0.09 |
| | PD | 0.443 (0.146 to 1.344) | 0.15 | 0.551 (0.127 to 2.385) | 0.43 |
| | Combined therapy | 1.309 (0.316 to 5.431) | 0.71 | 1.225 (0.188 to 7.966) | 0.83 |
| | CVD | 0.838 (0.370 to 1.897) | 0.67 | 0.663 (0.261 to 1.686) | 0.39 |
| | Diabetes | 1.072 (0.474 to 2.426) | 0.87 | 1.502 (0.555 to 4.065) | 0.42 |
| | BMI | 0.952 (0.876 to 1.036) | 0.26 | 0.931 (0.832 to 1.041) | 0.21 |
| | Hemoglobin | 1.150 (0.831 to 1.591) | 0.40 | 1.052 (0.715 to 1.548) | 0.80 |
| | Creatinine | 1.026 (0.901 to 1.169) | 0.70 | | |
| | Urea nitrogen | 0.988 (0.960 to 1.017) | 0.40 | | |
| | Serum albumin | 0.930 (0.380 to 2.275) | 0.87 | 0.816 (0.211 to 3.149) | 0.77 |
| | CRP * | 0.935 (0.651 to 1.342) | 0.71 | | |
| | β2 microglobulin | 1.021 (0.959 to 1.086) | 0.52 | | |
| | Use of ESAs | 0.607 (0.205 to 1.798) | 0.37 | | |
| Our original scale of fatigue | | | | | |
| | Age | 1.003 (0.969 to 1.038) | 0.87 | 0.953 (0.904 to 1.005) | 0.08 |
| | Male gender | 1.051 (0.464 to 2.384) | 0.90 | 1.222 (0.417 to 3.582) | 0.71 |
| | Dialysis duration * | 1.101 (0.821 to 1.476) | 0.52 | 1.129 (0.717 to 1.776) | 0.60 |

*(Continued)*

**Table 4.** (Continued)

|  |  | Unadjusted | | Adjusted | |
|---|---|---|---|---|---|
|  |  | Odds ratio (95% CI) | P value | Odds ratio (95% CI) | P value |
|  | Employed | 0.436 (0.196 to 0.970) | 0.04 | 0.440 (0.154 to 1.257) | 0.13 |
|  | Smoker | 0.157 (0.031 to 0.792) | 0.03 | 0.140 (0.021 to 0.944) | 0.04 |
|  | Dialysis modality |  |  |  |  |
|  | HD | Ref. |  | Ref. |  |
|  | Online HDF | 0.293 (0.086 to 1.003) | 0.051 | 0.267 (0.064 to 1.114) | 0.07 |
|  | PD | 0.309 (0.087 to 1.098) | 0.07 | 0.601 (0.114 to 3.154) | 0.55 |
|  | Combined therapy | 0.917 (0.170 to 4.930) | 0.92 | 0.958 (0.112 to 8.208) | 0.97 |
|  | CVD | 0.951 (0.419 to 2.157) | 0.90 | 0.655 (0.238 to 1.801) | 0.41 |
|  | Diabetes | 1.987 (0.829 to 4.760) | 0.12 | 2.475 (0.782 to 7.833) | 0.12 |
|  | BMI | 1.011 (0.931 to 1.098) | 0.79 | 0.946 (0.842 to 1.063) | 0.35 |
|  | Hemoglobin | 1.239 (0.885 to 1.735) | 0.21 | 0.922 (0.591 to 1.438) | 0.72 |
|  | Creatinine | 1.048 (0.918 to 1.197) | 0.49 |  |  |
|  | Urea nitrogen | 1.004 (0.977 to 1.033) | 0.76 |  |  |
|  | Serum albumin | 1.050 (0.426 to 2.587) | 0.92 | 0.916 (0.212 to 3.946) | 0.91 |
|  | CRP * | 1.276 (0.872 to 1.867) | 0.21 |  |  |
|  | β2 microglobulin | 1.035 (0.973 to 1.101) | 0.27 |  |  |
|  | Use of ESAs | 0.212 (0.046 to 0.986) | 0.048 | 0.195 (0.031 to 1.213) | 0.08 |

Abbreviations: CI, confidence interval; POMS, Profile of Mood States; HD, hemodialysis; HDF, hemodiafiltration; PD, peritoneal dialysis; CVD, cardiovascular disease; BMI, body mass index; CRP, C-reactive protein; ESAs, erythropoiesis-stimulating agents; VAS, Visual Analogue Scale.

* Dialysis duration and CRP are log-transformed.

possible that fatigue may have a negative impact patients' will to work. Interestingly, the prevalence of fatigue, defined as Grades 3, 4 and 5 in our original scale of fatigue, and the severity of fatigue, indicated by the scores of this scale, were significantly lower on dialysis-free days than on dialysis days in patients on HD and online HDF. Thus, these daily fluctuations in fatigue in relation to HD and online HDF session warrant further investigations.

The POMS, VAS, and our original scale of fatigue were used to evaluate fatigue in the present study. Although the 36-item Short-Form general health survey (SF-36), especially its vitality scale is frequently used in this area [2, 3], the POMS was used instead for two reasons. First, it includes the fatigue scale. Second, it is widely used in clinical practice and covered by the public medical insurance system in Japan. The scales of fatigue might differ by several sociodemographic factors such as age, sex, and race. The median scores of the fatigue subscale of the POMS scores in a Polish study including 115 dialysis patients were around 5 to 11, lower than in the present study [27], whereas this score at baseline of a randomized, controlled trial including 15 HD patients in the U.S.A. was similar to the present study [28]. The associations between the scores of these three scales were significant but weak. Additionally, the effects of several confounding factors, such as age, BMI, employment status and serum creatinine, on the presence or severity of fatigue differed depending on the scales. These discrepancies indicate the fact that different measures could yield different results. However, the presence or the severity of fatigue assessed using all scales did not differ significantly among patients receiving different dialysis modalities.

There were some limitations in this study. First, the cross-sectional, observational nature of the study allowed the identification of associations, but not causal relationships between fatigue and clinical variables, including dialysis modality. Additionally, the effect of residual

**Table 5. Multiple regression model for the scales of fatigue in patients without depression.**

| Variable | Unadjusted | | | | Adjusted | | | |
|---|---|---|---|---|---|---|---|---|
| | Regression coefficient | 95%CI | t value | p value | Regression coefficient | 95%CI | t value | p value |
| POMS (log-transformed) | | | | | | | | |
| Age | 0.0015 | -0.0037 to 0.0066 | 0.56 | 0.58 | -0.0030 | -0.0098 to 0.0039 | -0.86 | 0.39 |
| Male gender | -0.0692 | -0.1921 to 0.0538 | -1.11 | 0.27 | -0.0016 | -0.1460 to 0.1427 | -0.02 | 0.98 |
| Dialysis duration * | 0.0171 | -0.0268 to 0.0611 | 0.77 | 0.44 | -0.0054 | -0.0665 to 0.0558 | -0.17 | 0.86 |
| Employed | -0.1138 | -0.2257 to -0.0019 | -2.01 | 0.046 | -0.1117 | -0.2472 to 0.0239 | -1.63 | 0.11 |
| Smoker | -0.103 | -0.312 to 0.107 | -0.97 | 0.33 | | | | |
| Dialysis modality | | | | | | | | |
| HD | Ref. | | | | Ref. | | | |
| Online HDF | -0.0618 | -0.2205 to 0.0970 | -0.77 | 0.44 | -0.0818 | -0.2587 to 0.0951 | -0.92 | 0.36 |
| PD | -0.128 | -0.293 to 0.038 | -1.52 | 0.13 | -0.102 | -0.313 to 0.108 | -0.96 | 0.34 |
| Combined therapy | 0.0510 | -0.1581 to 0.2601 | 0.48 | 0.63 | 0.0648 | -0.2027 to 0.3322 | 0.48 | 0.63 |
| CVD | -0.0326 | -0.1560 to 0.0908 | -0.52 | 0.60 | -0.0750 | -0.2069 to 0.0569 | -1.13 | 0.26 |
| Diabetes | -0.0223 | -0.1458 to 0.1012 | -0.36 | 0.72 | 0.0178 | -0.1232 to 0.1587 | 0.25 | 0.80 |
| BMI | -0.0079 | -0.0201 to 0.0044 | -1.28 | 0.21 | -0.0090 | -0.0242 to 0.0062 | -1.17 | 0.24 |
| Hemoglobin | -0.0309 | -0.0794 to 0.0176 | -1.26 | 0.21 | -0.0391 | -0.0927 to 0.0145 | -1.45 | 0.15 |
| Creatinine | -0.00451 | -0.02416 to 0.01515 | -0.45 | 0.65 | | | | |
| Urea nitrogen | -0.00011 | -0.00429 to 0.00407 | -0.05 | 0.96 | | | | |
| Serum albumin | -0.0165 | -0.1530 to 0.1200 | -0.24 | 0.81 | -0.0249 | -0.2159 to 0.1661 | -0.26 | 0.80 |
| CRP * | 0.0036 | -0.0535 to 0.0608 | 0.13 | 0.90 | | | | |
| β2 microglobulin | 0.00693 | -0.00259 to 0.01644 | 1.45 | 0.15 | | | | |
| Use of ESAs | 0.0432 | -0.1237 to 0.2102 | 0.51 | 0.61 | | | | |
| VAS | | | | | | | | |
| Age | -0.081 | -0.434 to 0.273 | -0.45 | 0.65 | -0.393 | -0.858 to 0.071 | -1.68 | 0.10 |
| Male gender | 4.22 | -4.08 to 12.52 | 1.01 | 0.32 | 5.82 | -3.69 to 15.33 | 1.21 | 0.23 |
| Dialysis duration * | 1.31 | -1.65 to 4.27 | 0.88 | 0.38 | 1.22 | -2.83 to 5.27 | 0.6 | 0.55 |
| Employed | -5.07 | -12.66 to 2.53 | -1.32 | 0.19 | -6.78 | -15.77 to 2.22 | -1.49 | 0.14 |
| Smoker | -3.92 | -18.53 to 10.68 | -0.53 | 0.60 | | | | |
| Dialysis modality | | | | | | | | |
| HD | Ref. | | | | Ref. | | | |
| Online HDF | -5.77 | -16.24 to 4.71 | -1.09 | 0.28 | -6.21 | -17.85 to 5.43 | -1.06 | 0.29 |
| PD | -6.80 | -17.79 to 4.19 | -1.22 | 0.22 | -4.45 | -18.38 to 9.49 | -0.63 | 0.53 |
| Combined therapy | 8.29 | -5.82 to 22.40 | 1.16 | 0.25 | 7.29 | -10.32 to 24.90 | 0.82 | 0.41 |
| CVD | 2.20 | -6.05 to 10.44 | 0.53 | 0.60 | 0.82 | -7.93 to 9.57 | 0.19 | 0.85 |
| Diabetes | 4.22 | -4.09 to 12.52 | 1.01 | 0.32 | 6.81 | -2.52 to 16.14 | 1.45 | 0.15 |
| BMI | -0.333 | -1.153 to 0.486 | -0.81 | 0.42 | -0.828 | -1.830 to 0.175 | -1.64 | 0.11 |
| Hemoglobin | 2.33 | -0.91 to 5.58 | 1.42 | 0.16 | 1.07 | -2.48 to 4.62 | 0.6 | 0.55 |
| Creatinine | 0.595 | -0.728 to 1.919 | 0.89 | 0.38 | | | | |
| Urea nitrogen | -0.098 | -0.385 to 0.189 | -0.68 | 0.50 | | | | |
| Serum albumin | -0.76 | -9.87 to 8.36 | -0.16 | 0.87 | -1.19 | -13.76 to 11.38 | -0.19 | 0.85 |
| CRP * | 0.04 | -3.64 to 3.72 | 0.02 | 0.98 | | | | |
| β2 microglobulin | 0.120 | -0.507 to 0.747 | 0.38 | 0.70 | | | | |
| Use of ESAs | 0.01 | -11.14 to 11.16 | 0.00 | 1.00 | | | | |
| Our original scale of fatigue | | | | | | | | |
| Age | 0.00967 | -0.00787 to 0.02720 | 1.09 | 0.28 | -0.0117 | -0.0339 to 0.0105 | -1.05 | 0.30 |
| Male gender | -0.105 | -0.523 to 0.314 | -0.49 | 0.62 | 0.100 | -0.360 to 0.560 | 0.43 | 0.67 |

*(Continued)*

**Table 5.** (Continued)

| Variable | | Unadjusted | | | | Adjusted | | | |
|---|---|---|---|---|---|---|---|---|---|
| | | Regression coefficient | 95%CI | t value | p value | Regression coefficient | 95%CI | t value | p value |
| | Dialysis duration * | 0.0392 | -0.1111 to 0.1894 | 0.52 | 0.61 | 0.0968 | -0.0981 to 0.2916 | 0.98 | 0.33 |
| | Employed | -0.521 | -0.895 to -0.146 | -2.75 | 0.01 | -0.408 | -0.844 to 0.028 | -1.86 | 0.07 |
| | Smoker | -0.955 | -1.641 to -0.269 | -2.76 | 0.01 | -0.803 | -1.510 to -0.096 | -2.25 | 0.03 |
| | Dialysis modality | | | | | | | | |
| | HD | Ref. | | | | | | | |
| | Online HDF | -0.420 | -0.944 to 0.104 | -1.59 | 0.12 | -0.422 | -0.990 to 0.146 | -1.47 | 0.14 |
| | PD | -0.311 | -0.858 to 0.237 | -1.12 | 0.26 | -0.107 | -0.783 to 0.570 | -0.31 | 0.76 |
| | Combined therapy | 0.543 | -0.1478 to 1.234 | 1.56 | 0.12 | 0.278 | -0.590 to 1.145 | 0.64 | 0.53 |
| | CVD | -0.103 | -0.522 to 0.316 | -0.49 | 0.63 | -0.323 | -0.744 to 0.098 | -1.52 | 0.13 |
| | Diabetes | 0.395 | -0.018 to 0.809 | 1.89 | 0.06 | 0.457 | -0.001 to 0.914 | 1.98 | 0.05 |
| | BMI | 0.0123 | -0.0294 to 0.0541 | 0.59 | 0.56 | -0.0064 | -0.0548 to 0.0420 | -0.26 | 0.79 |
| | Hemoglobin | 0.0224 | -0.1434 to 0.1882 | 0.27 | 0.79 | -0.0830 | -0.2561 to 0.0901 | -0.95 | 0.34 |
| | Creatinine | -0.0013 | -0.0681 to 0.0654 | -0.04 | 0.97 | | | | |
| | Urea nitrogen | -0.01070 | -0.02475 to 0.00335 | -1.51 | 0.13 | | | | |
| | Serum albumin | -0.353 | -0.804 to 0.098 | -1.55 | 0.12 | -0.229 | -0.849 to 0.390 | -0.73 | 0.46 |
| | CRP * | 0.113 | -0.067 to 0.293 | 1.25 | 0.22 | | | | |
| | β2 microglobulin | 0.0163 | -0.0153 to 0.0479 | 1.03 | 0.31 | | | | |
| | Use of ESAs | -0.350 | -0.913 to 0.214 | -1.23 | 0.22 | | | | |

Abbreviations: CI, confidence interval; POMS, Profile of Mood States; HD, hemodialysis; HDF, hemodiafiltration; PD, peritoneal dialysis; CVD, cardiovascular disease; BMI, body mass index; CRP, C-reactive protein; ESAs, erythropoiesis-stimulating agents; VAS, Visual Analogue Scale.

* Dialysis duration and CRP are log-transformed.

confounding factors including adequacy or frequency of dialysis and residual renal function, could not assessed. Second, assessment of inflammation, an important factor related to fatigue, was difficult, because of missing values and lower levels, including below the detectable level, of CRP. Third, selection bias, especially physicians' preference, for choice of dialysis modalities was considerable, because this study was not a randomized, controlled study, but an observational study. Fourth, although the results of our original scale of fatigue were significantly associated with those of the POMS and VAS, convergent and discriminant validity, test-retest reliability, or social desirability could not be clarified using this study design. Fifth, the details of the screening process at each center were not known. Further selection bias may have been introduced by recruiting predominantly patients who were able to read and respond to the questionnaires themselves, thereby compromising the generalizability of the study. Sixth, since patients receiving combined therapy with PD and HD answered all self-reported questionnaires not on the HD session days but at regular outpatient visits, the impact of HD on the questionnaire scores was unclear, because the timing of assessment related to HD sessions differed among patients in this group. Seventh, sample size was small, which may be the reason for not detecting a statistically significant difference in fatigue levels across modalities.

## Conclusions

The results indicated that the type of dialysis modality, including HD, online HDF, PD and combined therapy with PD and HD did not appear to be associated with the prevalence or severity of fatigue of patients on dialysis. In patients undergoing HD or online HDF, fatigue

levels were significantly lower on dialysis-free days. Further research to explore the differences in pathophysiology of fatigue between dialysis modalities may be warranted.

## Supporting information

**S1 Data. Original data set.**
(ZIP)

## Acknowledgments

The authors gratefully acknowledge the support and participation of the patients in Shinagawa Kidney Clinic (Tokyo, Japan), Iidabashi Murai Clinic (Tokyo, Japan), Aoto Kidney Clinic (Tokyo, Japan), and Shin Kashiwa Clinic (Kashiwa, Chiba)

## Author Contributions

**Conceptualization:** Yukio Maruyama, Masaaki Nakayama, Atsushi Ueda, Mariko Miyazaki, Takashi Yokoo.

**Formal analysis:** Yukio Maruyama, Masaaki Nakayama, Atsushi Ueda.

**Funding acquisition:** Masaaki Nakayama, Atsushi Ueda.

**Investigation:** Yukio Maruyama, Masaaki Nakayama, Atsushi Ueda.

**Methodology:** Yukio Maruyama, Masaaki Nakayama, Atsushi Ueda.

**Supervision:** Mariko Miyazaki, Takashi Yokoo.

**Writing – original draft:** Yukio Maruyama.

**Writing – review & editing:** Yukio Maruyama.

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
