## [Decision Letter · Decision Letter 0]

23 Jul 2020

PONE-D-20-17573

Comparisons of fatigue between dialysis modalities: A cross-sectional study

PLOS ONE

Dear Dr. Maruyama,

Thank you for submitting your manuscript to PLOS ONE. After careful consideration, we feel that it has merit but does not fully meet PLOS ONE’s publication criteria as it currently stands and will need major revisions in order to be considered for publication. Therefore, we invite you to submit a revised version of the manuscript that addresses the points raised during the review process.

In particular, the aims of this study should be clarified (validation study versus assessment of fatigue accross different dialysis modality), the use of POMS instead of SF-36 needs further justification and more emphasis should be given to the difficulties (and limitations) of interpreting results when using different measures of fatigue as presented in this study. The discussion should focus more on the outcome of fatigue then interventions to improve fatigue. 

We look forward to receiving your revised manuscript.

Kind regards,

Andrea K. Viecelli

Academic Editor

PLOS ONE

Journal Requirements:

2.Thank you for stating the following financial disclosure:

 [This study was supported in part by research grants from Baxter International, Inc. (number 16CECPDAP0002). The funders had no role in study design, data collection and analysis, decision to publish, or preparation of the manuscript.].              

3.Thank you for stating the following in the Competing Interests section:

[I have read the journal's policy and the authors of this manuscript have the following competing interests: Y.M. and M.N. have received scholarship funds from Baxter International, Inc. and Terumo Corporation. No other authors have any conflicts of interest to declare.].

Reviewers' comments:

Reviewer's Responses to Questions

**Comments to the Author**

1. Is the manuscript technically sound, and do the data support the conclusions?

Reviewer #1: Yes

Reviewer #2: Partly

2. Has the statistical analysis been performed appropriately and rigorously? 

Reviewer #1: Yes

Reviewer #2: Yes

3. Have the authors made all data underlying the findings in their manuscript fully available?

Reviewer #1: No

Reviewer #2: Yes

4. Is the manuscript presented in an intelligible fashion and written in standard English?

Reviewer #1: Yes

Reviewer #2: No

5. Review Comments to the Author

Reviewer #1: I read with interest the manuscript “Comparisons of fatigue between dialysis modalities: A cross-sectional study” by Maruyama and colleagues. This study provides interesting findings on fatigue in different dialysis modalities, as assessed through three different fatigue evaluation scales (POMS, VAS and an original scale).

Nevertheless, I have a few comments and questions to ask the authors. For simplicity, I will comment by sections.

METHODS

Subjects

1. Could the authors comment on how the patients were screened for inclusion in the study? Were all patients in the included centers screened for inclusion? Was there a minimum duration of dialysis for inclusion (eg. established on dialysis for at least 3 months)?

2. The authors specify that “any patient that had a diagnosable mental health disorder or medical illness, including dementia and terminal cancer were excluded”. As patients with depression, CVD, diabetes, etc. are included in this study, it appears that the “diagnosable mental health disorder or medical illness” warranting exclusion from the study were more specific. The authors should further specify the exact exclusions and/or modify this statement in the manuscript.

Assessments of fatigue and depression

3. All used scales are very well described in the manuscript. However, no details are provided on how the responses to the self-reported questionnaires/scales were collected: were the patients completing the questionnaires in a written form by themselves or verbally answering to the questions to a member of staff? The authors should provide more details on the exact “self-reporting” procedure and if any exclusion from the study would have been related to these. For example, if the questionnaires were completed in a written format, were patients unable to write/read excluded or offered any help to complete the questionnaire?

Statistical analysis

4. Were the variables included in the multivariate analysis chosen on biological/clinical plausibility alone or was a univariate analysis done? As smoking was significantly different across groups, can the authors explain why it was not included in the multivariate analysis?

RESULTS

5. Following on question #1, the authors should provide some information on the included cohort compared to the dialysis population in the participating centers. How many patients were screened for inclusion; how many were excluded and for which reason, etc? This might also need to be mentioned in the limitations of the study depending on the representativity of this cohort compared to the dialysis population screened.

6. Can the authors comment on the predominantly male cohort: is this proportion representative of the dialysis population in Japan or is this related to higher consent to study by men or exclusion of more women due to exclusion criteria?

7. Could the authors provide a brief statement on modality choice in the participating centers (HD vs HDF vs PD vs combined HD/PD): patients’ choice, medical indication, ‘PD first’ policy, etc?

8. In the introduction, the authors mention that “It is unclear whether dialysis-related factors such as the type of dialysis modality and the adequacy or frequency of dialysis are associated with fatigue”. Were data on dialysis prescription (number of exchanges and volume of PD; treatment time and frequency on HD, etc) and on dialysis adequacy (Kt/v or URR) collected as part of this study? If so, these should be described and potentially included in the regression models. If these data were not collected, it should be mentioned in the limitation of the study as they could potentially be important confounders.

9. In the paragraph describing results included in Tables 2 and 3 (p.10-11), the same sentences appear twice. Table 2 describes the logistic regression model for highest score of fatigue (dichotomic outcome) and not the severity of fatigue. This should be corrected in the manuscript. In the second part of that paragraph (describing Table 3), the same sentence describing the association of employment status is used, mentioning the “lower risk of fatigue”. Here, it appears that this results should refer to a “lower severity/score of fatigue” (continuous outcome).

DISCUSSION

10. Could the authors comment on the median VAS and POMS values obtained in their study compared to previous studies?

For example, in a Malaysian study comparing quality of life in HD vs CAPD, mean VAS scores were around 76-77 vs. 48 in the present study.

[Surendra NK, Abdul Manaf MR, Hooi LS, et al. Health related quality of life of dialysis patients in Malaysia: Haemodialysis versus continuous ambulatory peritoneal dialysis. BMC Nephrol. 2019;20(1):151. Published 2019 Apr 30. doi:10.1186/s12882-019-1326-x]

In contrast, a Polish study evaluating defense mechanism in dialysis patients found the fatigue subscale of the POMS around 5-11 vs. 11.3 in the present study.

[Nowak Z, Wańkowicz Z, Laudanski K. Denial Defense Mechanism in Dialyzed Patients. Med Sci Monit. 2015;21:1798-1805. Published 2015 Jun 22. doi:10.12659/MSM.893331]

The generalizability of the results from the present study should be mentioned in the discussion in light of such differences or similarities.

FIGURE 1

11. In Figure 1, for the graph representing the results from the original scale, asterixis (*) are used but no legend for this is provided.

In conclusion, this manuscript presents very interesting findings on severity of fatigue in different dialysis modality, including the combined therapy of PD and HD, which is used much more frequently in Japan than elsewhere in the world. The cross-sectional, observational nature of the study, the potential for selection bias and residual confounding are limitations of this study, which have been clearly identified by the authors in the discussion.

Please note that the data underlying the findings is described as fully available within the manuscript and its Supporting Information files on the submission but no supporting files were attached to the manuscript for review so I cannot comment on the “full availability” of the data.

Reviewer #2: While fatigue is very important for patients on dialysis, I'm not sure that this study adds anything new to the existing body of literature on this topic. Furthermore, I have outlined some major methodological concerns below.

Aim - It is unclear whether the aim is to investigate the levels of fatigue across dialysis modalities or validate a new fatigue measure the authors have created

Introduction:

- Second last paragraph: QOL and fatigue are not interchangeable so this paragraph seems a bit random – either explain how fatigue impacts QOL and thus is the research question for this study, or explore literature around fatigue instead

- Last paragraph: why would comparing modalities expected to help identify which subjects will benefit from interventions to reduce fatigue?

Methods:

- I think the authors should justify why POMS was chosen for this study. If frequency of use was the basis, why wasn’t SF-36 used? POMS includes a subscale for fatigue but given that it is more of a symptom checklist-type measure with a focus on ‘moods’ than a symptom-specific questionnaire, it seems an odd choice of measure to which the authors’ measure is compared

- What was the sampling methodology?

- The readers would benefit from greater details about the ‘original measure’. How was it developed/through what process? Has the content validity been established? Is this the first time that it has been used in a study? Why/how were grades 3, 4 and 5 chosen as ‘fatigued states’? Is this measure described else where in the literature? If so, please add in the relevant citation

- Depression needs to be better integrated into the rationale of this study

Results:

- Given that POMS, VAS and the original scale were all used to explore fatigue, the rho seems quite low. explore this in the discussion

- POMS and VAS did not differ across four groups, but the original scale did – what does this tell us about the fatigue levels across these groups? Or what does this say about the validity of the original scale?

Discussion

- Discrepancy in results in existing literature regarding fatigue may also be due to the fact that measures of fatigue vary across these studies (as evident in this study where different measures yield different results)

- Discussion on the interventions for fatigue seems out of place – link the results of this study back to what the implications are for future research aiming to look at interventions for fatigue across different modalities

- 4th limitation is noted as the fact that ‘convergent and discriminant validity, test-retest reliability or social desirability could not be clarified using this study design’. This is a major flaw of this study, as using a measure that has not been adequately validated runs the risk of yielding not only inconclusive but also misleading results. I would advise that the authors rethink the true aim of this study

6. PLOS authors have the option to publish the peer review history of their article (what does this mean?). If published, this will include your full peer review and any attached files.

Reviewer #1: No

Reviewer #2: No

---

## [Author Response · Author response to Decision Letter 0]

2 Sep 2020

Response to Reviewers

In particular, the aims of this study should be clarified (validation study versus assessment of fatigue across different dialysis modality), the use of POMS instead of SF-36 needs further justification and more emphasis should be given to the difficulties (and limitations) of interpreting results when using different measures of fatigue as presented in this study. The discussion should focus more on the outcome of fatigue then interventions to improve fatigue. 

RESPONSE: Thank you for raising these very important points. We have clarified the aims of this study in the Introduction as follows: “The aim of this cross-sectional study was to compare the prevalence or severity of fatigue among patients receiving different dialysis modalities, including HD, online HDF, PD, and combined therapy with PD and HD.” As we mentioned in the Limitation, we could not validate the effectiveness of our original scale of fatigue using this study design. Since the last sentence of the Introduction of "Additionally, the effectiveness of our original, very simple scale of fatigue was verified" is inadequate, we have deleted this sentence. We did not use SF-36, especially its the vitality scale, despite the frequently use in this area [Ref. 2, Ref. 3] and used POMS instead for two reasons. First, it includes the fatigue scale. Second, it is widely used in clinical practice and covered by the public medical insurance system in Japan. The associations between the scores of POMS, VAS and our original scale of fatigue were significant but weak. Additionally, the effects of several confounding factors, such as age, BMI, employment status and serum creatinine, on the presence or severity of fatigue differed depending on the scales. These discrepancies indicate the fact that different measures could yield different results. We have added this information to the Discussion. As suggested, we have deleted the discussion on the interventions for fatigue because this information is not necessary. We modified the discussion focused more on the outcome of fatigue. 

RESPONSE: As suggested, we amended Funding Statement within our cover letter.

RESPONSE: As suggested, we updated Competing Interests statement in our cover letter.

Reviewer #1: I read with interest the manuscript “Comparisons of fatigue between dialysis modalities: A cross-sectional study” by Maruyama and colleagues. This study provides interesting findings on fatigue in different dialysis modalities, as assessed through three different fatigue evaluation scales (POMS, VAS and an original scale).

Nevertheless, I have a few comments and questions to ask the authors. For simplicity, I will comment by sections.

METHODS

Subjects

1. Could the authors comment on how the patients were screened for inclusion in the study? Were all patients in the included centers screened for inclusion? Was there a minimum duration of dialysis for inclusion (eg. established on dialysis for at least 3 months)?

RESPONSE: Thank you for raising this important point. As we described in the Materials and methods, we excluded patients under 18 years of age, in addition to patients with a mental disorder, such as schizophrenia, depression or dementia, and those with terminal cancer. We did not determine other inclusion or exclusion criteria and did not exclude patients with shorter duration of dialysis. Unfortunately, we are unaware of the detail of screening process at each center. We have added this information to the Limitation.

2. The authors specify that “any patient that had a diagnosable mental health disorder or medical illness, including dementia and terminal cancer were excluded”. As patients with depression, CVD, diabetes, etc. are included in this study, it appears that the “diagnosable mental health disorder or medical illness” warranting exclusion from the study were more specific. The authors should further specify the exact exclusions and/or modify this statement in the manuscript.

RESPONSE: Thank you for pointing out this error. As suggested, we did not exclude patients with history of CVD or diabetes. Therefore, the word of “medical illness” is inappropriate. We have corrected the description of exclusion criteria as follows: “Patients under 18 years of age, in addition to patients with a mental disorder, such as schizophrenia, depression or dementia, and those with terminal cancer were excluded.” 

Assessments of fatigue and depression

3. All used scales are very well described in the manuscript. However, no details are provided on how the responses to the self-reported questionnaires/scales were collected: were the patients completing the questionnaires in a written form by themselves or verbally answering to the questions to a member of staff? The authors should provide more details on the exact “self-reporting” procedure and if any exclusion from the study would have been related to these. For example, if the questionnaires were completed in a written format, were patients unable to write/read excluded or offered any help to complete the questionnaire?

RESPONSE: Thank you for noticing this issue. We confirmed that all participants read and wrote questionnaires for themselves, although we did not excluded patients who need help to answer “written format” questionnaires. We noticed that there was a bias in recruiting participants for this study. We have added this information to the Materials and methods, the Results and the Limitation.

Statistical analysis

4. Were the variables included in the multivariate analysis chosen on biological/clinical plausibility alone or was a univariate analysis done? As smoking was significantly different across groups, can the authors explain why it was not included in the multivariate analysis?

RESPONSE: Thank you for raising this important point. On multivariate analysis, we selected several confounding factors reported to be associated with fatigue in dialyzed patients. However, there could be a possible confounding factor, including smoking status, as the reviewer suggested. Therefore, we changed the methods of variable selection. We also included confounding factor that had shown an appreciable association (P<0.1) with a measure of fatigue in multivariate analysis in order to avoid omissions error and created new Table 2 to 5. We have added this information to the Materials and methods and have modified the descriptions in the Results and Discussion, because the results of regression analysis were partially changed.

RESULTS

5. Following on question #1, the authors should provide some information on the included cohort compared to the dialysis population in the participating centers. How many patients were screened for inclusion; how many were excluded and for which reason, etc? This might also need to be mentioned in the limitations of the study depending on the representativity of this cohort compared to the dialysis population screened.

RESPONSE: Thank you for raising this important point. As described before, we are unaware of the detail of screening process at each center. Additionally, there could be a bias in recruiting participants for this study and representative sampling may not reflect the general dialysis population, because all participants read and wrote questionnaires for themselves although we did not excluded patients who need help to answer written format questionnaires. We have added this information to the Limitation.

6. Can the authors comment on the predominantly male cohort: is this proportion representative of the dialysis population in Japan or is this related to higher consent to study by men or exclusion of more women due to exclusion criteria?

RESPONSE: Thank you for this helpful comment. The percentage of male participants in this study was 69.1% and we think that it is similar to the general dialysis population in Japan. According to annual surveys of dialysis facilities throughout Japan in 2017, 65% of prevalent dialysis patients were male [Ref. 19].

7. Could the authors provide a brief statement on modality choice in the participating centers (HD vs HDF vs PD vs combined HD/PD): patients’ choice, medical indication, ‘PD first’ policy, etc?

RESPONSE: Thank you for this helpful comment. As we described in the Limitation, selection bias, especially physicians’ preference, for choice of dialysis modalities was considerable, because this study was not a randomized controlled study, but an observational study. PD First policy is recommended in order to maximize the advantages of PD therapy including excellent comparable survival, lower cost and improved quality of life also in Japan [Ref. 20, Ref. 21]. However, its long-term continuation is difficult because of a decline in residual renal function and deterioration of the peritoneal membrane. Patients who become unmanageable only by PD need to change the modality of RRT. Although they often directly transfer to HD, transfer to combined therapy with PD and HD, generally consists of 5-6 days of PD and commonly one HD session per week, was relatively common in Japan. PD therapy and combined therapy with PD and HD are thought to be performed especially among young and socially engaged patients. Indeed, the mean age of PD patients was 63.5 years, whereas the mean age of HD patients was 69.4 years [Ref. 19]. Additionally, the number of patients treated with online HDF have been increasing after the 2012 revision to the medical reimbursement system, and have reached 70000, approximately 30% of all dialysis patients at the end of 2017 in Japan. Since the mean age of HDF patients was 66.6 years and approximately 3 years younger than HD patients, this therapy is also thought to be performed especially among young and socially engaged patients [Ref. 19]. Therefore, we think that the prevalence or the severity of fatigue should be investigated not only among patients on PD and HD but those on online HDF and combined therapy with PD and HD. We have added this information to the Introduction and the Discussion. 

8. In the introduction, the authors mention that “It is unclear whether dialysis-related factors such as the type of dialysis modality and the adequacy or frequency of dialysis are associated with fatigue”. Were data on dialysis prescription (number of exchanges and volume of PD; treatment time and frequency on HD, etc) and on dialysis adequacy (Kt/v or URR) collected as part of this study? If so, these should be described and potentially included in the regression models. If these data were not collected, it should be mentioned in the limitation of the study as they could potentially be important confounders.

RESPONSE: Thank you for noticing this issue. As the reviewer indicated, we did not have the data of neither adequacy nor frequency of dialysis. We have deleted these descriptions from the Introduction. Furthermore, we have added following sentences to the Limitations: “The effect of residual confounding factors including adequacy or frequency of dialysis and residual renal function, was unavoidable.”

9. In the paragraph describing results included in Tables 2 and 3 (p.10-11), the same sentences appear twice. Table 2 describes the logistic regression model for highest score of fatigue (dichotomic outcome) and not the severity of fatigue. This should be corrected in the manuscript. In the second part of that paragraph (describing Table 3), the same sentence describing the association of employment status is used, mentioning the “lower risk of fatigue”. Here, it appears that this results should refer to a “lower severity/score of fatigue” (continuous outcome).

RESPONSE: Thank you for this helpful comment. As suggested, we have modified the description of the Results. In the results of logistic regression analysis, we changed the description of outcome from the severity of fatigue to the presence of fatigue, because objective variables were not continuous but dichotomic variables. In addition, we changed the description of outcome from the risk of fatigue to the severity of fatigue in multiple regression analysis. 

DISCUSSION

10. Could the authors comment on the median VAS and POMS values obtained in their study compared to previous studies?

For example, in a Malaysian study comparing quality of life in HD vs CAPD, mean VAS scores were around 76-77 vs. 48 in the present study.

[Surendra NK, Abdul Manaf MR, Hooi LS, et al. Health related quality of life of dialysis patients in Malaysia: Haemodialysis versus continuous ambulatory peritoneal dialysis. BMC Nephrol. 2019;20(1):151. Published 2019 Apr 30. doi:10.1186/s12882-019-1326-x]

In contrast, a Polish study evaluating defense mechanism in dialysis patients found the fatigue subscale of the POMS around 5-11 vs. 11.3 in the present study.

[Nowak Z, Wańkowicz Z, Laudanski K. Denial Defense Mechanism in Dialyzed Patients. Med Sci Monit. 2015;21:1798-1805. Published 2015 Jun 22. doi:10.12659/MSM.893331]

The generalizability of the results from the present study should be mentioned in the discussion in light of such differences or similarities.

RESPONSE: Thank you for this helpful comment. The scales of fatigue might differ from several sociodemographic factors such as age, gender and race. Unfortunately, we could not find the report comparing the scores of these scales between different populations. The VAS was graduated from 0 (worst imaginable health state) to 100 (best imaginable health state) in the Malaysian study the reviewer introduced, whereas it was graduated from 0 (no tiredness at all) to 100 (complete exhaustion) in our study. As indicated, the fatigue subscale of the POMS scores in the Polish study including 115 dialysis patients the reviewer introduced were lower than those of our study [Ref. 27]. On the contrary, these scores at baseline of randomized controlled trial including 15 HD patients in U.S. were approximately similar to the present study [Ref. 28]. We have added this information to the Discussion. 

FIGURE 1

11. In Figure 1, for the graph representing the results from the original scale, asterixis (*) are used but no legend for this is provided.

RESPONSE: As suggested, we have added the explanation of asterixis to the Figure Legend.

In conclusion, this manuscript presents very interesting findings on severity of fatigue in different dialysis modality, including the combined therapy of PD and HD, which is used much more frequently in Japan than elsewhere in the world. The cross-sectional, observational nature of the study, the potential for selection bias and residual confounding are limitations of this study, which have been clearly identified by the authors in the discussion.

RESPONSE: Thank you for the favorable review you gave our manuscript.

Please note that the data underlying the findings is described as fully available within the manuscript and its Supporting Information files on the submission but no supporting files were attached to the manuscript for review so I cannot comment on the “full availability” of the data.

RESPONSE: As suggested, we attached data set we used to the revised manuscript. 

Reviewer #2: While fatigue is very important for patients on dialysis, I'm not sure that this study adds anything new to the existing body of literature on this topic. Furthermore, I have outlined some major methodological concerns below.

RESPONSE: Thank you for the review you gave our manuscript. We wholly revised our manuscript according a number of helpful comments. As described before, the aim of this cross-sectional study was to compare the prevalence or severity of fatigue among patients receiving different dialysis modalities, including HD, online HDF, PD, and combined therapy with PD and HD. The main result was the prevalence and the severity of fatigue was not significantly different among patients receiving different dialysis modalities. The present study is the first study examined fatigue condition among patients receiving online HDF or combined therapy with PD and HD. We have added this information to the Introduction and Discussion.

Aim - It is unclear whether the aim is to investigate the levels of fatigue across dialysis modalities or validate a new fatigue measure the authors have created

RESPONSE: Thank you for raising this important point. We have clarified the aims of this study in the Introduction as follows: “The aim of this cross-sectional study was to compare the prevalence or severity of fatigue among patients receiving different dialysis modalities, including HD, online HDF, PD, and combined therapy with PD and HD.” As we mentioned in the Limitation, we could not validate the effectiveness of our original scale of fatigue using this study design. Since the description of "Additionally, the effectiveness of our original, very simple scale of fatigue was verified" is inadequate, we have deleted this sentence.

Introduction:

- Second last paragraph: QOL and fatigue are not interchangeable so this paragraph seems a bit random – either explain how fatigue impacts QOL and thus is the research question for this study, or explore literature around fatigue instead

RESPONSE: Thank you for raising this important point. We completely agree with the point that QOL and fatigue are not compatible. As suggested, we have deleted the description regarding the comparisons of QOL between different dialysis modalities. Instead, we focused on the clinical condition of fatigue. 

- Last paragraph: why would comparing modalities expected to help identify which subjects will benefit from interventions to reduce fatigue?

RESPONSE: Thank you for this helpful comment. As suggested, we could not identify the subjects having benefit of interventions to reduce fatigue from this study design. We have deleted this sentence.

Methods:

- I think the authors should justify why POMS was chosen for this study. If frequency of use was the basis, why wasn’t SF-36 used? POMS includes a subscale for fatigue but given that it is more of a symptom checklist-type measure with a focus on ‘moods’ than a symptom-specific questionnaire, it seems an odd choice of measure to which the authors’ measure is compared

RESPONSE: Thank you for raising this important point. As mentioned before, we did not use SF-36, especially its the vitality scale, despite the frequently use in this area [Ref. 2, Ref. 3] and used POMS instead for two reasons. First, it includes the fatigue scale. Second, it is widely used in clinical practice and covered by the public medical insurance system in Japan. We have added this information to the Discussion. 

- What was the sampling methodology?

RESPONSE: Thank you for this helpful comment. As mentioned before, we decided exclusion criteria, however we are unaware of the detail of screening process at each center. We have added following sentences to the Limitations: “We are unaware of the detail of screening process at each center. Additionally, there could be a bias in recruiting participants for this study and representative sampling may not reflect the general dialysis population, because all participants read and wrote questionnaires for themselves although we did not excluded patients who need help to answer written format questionnaires.” 

- The readers would benefit from greater details about the ‘original measure’. How was it developed/through what process? Has the content validity been established? Is this the first time that it has been used in a study? Why/how were grades 3, 4 and 5 chosen as ‘fatigued states’? Is this measure described else where in the literature? If so, please add in the relevant citation

RESPONSE: Thank you for raising this important point. As we mentioned in the Materials and methods, we developed our original scale of fatigue and have already used in the previous two studies (Ref.14 and Ref. 15). However, only this scale was used for the assessment of fatigue in these two studies. It is first time to compare the findings of this scale with those of well-established scales and we found that the scores of the POMS, VAS, and our original scale of fatigue were significantly inter-related. However, convergent and discriminant validity, test-retest reliability, or social desirability could not be clarified using this study design as we mentioned in the Limitation. As described in the Materials and methods, our original scale of fatigue is a five-grade evaluation including Grade 1 (inexhaustible, feels well, and extremely active), Grade 2 (tireless, acts in the ordinary way without a sense of fatigue), Grade 3 (mild fatigue, acts in the ordinary way but feels tired), Grade 4 (moderate fatigue, feels tired with light work), and Grade 5 (intense fatigue, very tired, and falls asleep). Therefore, we defined Grades 3 to 5 as fatigued state because the examinee realizes the presence of mild, moderate or intense fatigue.

- Depression needs to be better integrated into the rationale of this study

RESPONSE: Thank you for this helpful comment. As we mentioned in the manuscript, it is well-known that depression, the most important psychological factor, is strongly associated with fatigue and they are closely interrelated. Although exclusion criteria included the patients with depression, we could not exclude the effects of depressive status on the prevalence or severity of fatigue. Therefore, we assessed the presence and severity of depressive symptoms using the BDI–II and conducted sensitivity analyses including only patients with lower score of BDI-II. Since the results of these sensitivity analysis were similar to the results including all subjects, depressive state was thought to have a low impact on the prevalence or severity of fatigue in our study. We have added this information to the Results and Discussion. 

Results:

- Given that POMS, VAS and the original scale were all used to explore fatigue, the rho seems quite low. explore this in the discussion

RESPONSE: Thank you for this helpful comment. As suggested, the associations between the scores of POMS, VAS and our original scale of fatigue were significant but weak. Additionally, the effects of several confounding factors, such as age, BMI, employment status and serum creatinine, on the presence or severity of fatigue differed depending on the scales. These discrepancies indicate the fact that different measures could yield different results. We have added this information to the Discussion. 

- POMS and VAS did not differ across four groups, but the original scale did – what does this tell us about the fatigue levels across these groups? Or what does this say about the validity of the original scale?

RESPONSE: Thank you for this helpful comment. Although only the score of our original scale of fatigue was significantly different among patients on different dialysis modalities in univariate analysis, this finding disappeared in multivariate analysis. Therefore, we considered that the severity of fatigue assessed by our original scale of fatigue were not different. As mentioned before, we could not validate the effectiveness of our original scale of fatigue using this study design. 

Discussion

- Discrepancy in results in existing literature regarding fatigue may also be due to the fact that measures of fatigue vary across these studies (as evident in this study where different measures yield different results)

RESPONSE: Thank you for this helpful comment. As suggested, the effects of several confounding factors, such as age, BMI, employment status and serum creatinine, on the presence or severity of fatigue differed depending on the scales. Additionally, the associations between the scores of POMS, VAS and our original scale of fatigue were significant but weak. These discrepancies indicate the fact that different measures could yield different results. We have added this information to the Discussion. 

- Discussion on the interventions for fatigue seems out of place – link the results of this study back to what the implications are for future research aiming to look at interventions for fatigue across different modalities

RESPONSE: Thank you for this helpful comment. We completely agree with the opinion that discussion on the interventions for fatigue was not necessary. As the Editor also pointed, we deleted the description of this discussion and focused more on the outcome of fatigue.

- 4th limitation is noted as the fact that ‘convergent and discriminant validity, test-retest reliability or social desirability could not be clarified using this study design’. This is a major flaw of this study, as using a measure that has not been adequately validated runs the risk of yielding not only inconclusive but also misleading results. I would advise that the authors rethink the true aim of this study

RESPONSE: Thank you for raising this important point. As mentioned before, the aim of our study was to compare the prevalence or severity of fatigue among patients receiving different dialysis modalities, including HD, online HDF, PD, and combined therapy with PD and HD. We could not validate the effectiveness of our original scale of fatigue using this study design. We have clarified the aims of this study in the Introduction. In this study, we employed our original scale of fatigue as a complementary test for direct patient-reported outcome, because it is easy to use, and its simplicity free usage. We used this test both dialysis and dialysis-free day and found that there were significant differences. The result may indicate the different diurnal and weekly profiles of fatigue between HD and PD, despite the equivalent levels of fatigue on dialysis day in PD and HD. We added this point in the revised manuscript, and we think this needs to be clarified in future study by using POMS, VAS and SF-36.

---

## [Decision Letter · Decision Letter 1]

22 Oct 2020

PONE-D-20-17573R1

Comparisons of fatigue between dialysis modalities: A cross-sectional study

PLOS ONE

Dear Dr. Maruyama,

Thank you for submitting your manuscript to PLOS ONE. After careful consideration, we feel that it has merit but does not fully meet PLOS ONE’s publication criteria as it currently stands. Therefore, we invite you to submit a revised version of the manuscript that addresses the points raised during the review process.

The authors have addressed previous comments. However, the manuscript needs significant editing to comply with standard English. There are too many typographical or grammatical errors to list here. The manuscript is not publishable in the current state and will require major editing and revisions. The original data set referred to under “supporting information” was not available to the editor or reviewer and will need to be accessible in order to comply with publication criteria. In addition to the reviewer’s comment, there are additional issues identified that require clarification:

Abstract:

Please avoid vague statements in the abstract like “The scores of the POMS, VAS, and our original scale of fatigue were weakly but significantly inter-related”. The actual results would be more helpful.  

“In the fatigue scales of on dialysis day by 3 tests, there were no statistical differences by modalities.” The meaning of this sentence is unclear. Please clarify and revise.

“The similar result was obtained in 122 patients without depression.” “The” should be removed.

The conclusion of the abstract and manuscript is unclear and needs rephrasing. What do the authors mean with “The results indicated the least impact of dialysis modalities on fatigue in dialysis patients including HD, online HDF, PD and combined therapy with PD and HD.”?

Methods: “All participants read and wrote questionnaires for themselves”. Participants presumably did not write the questionnaires but responded to the questions without assistance.

The references need reformatting.

We look forward to receiving your revised manuscript.

Kind regards,

Andrea K. Viecelli

Academic Editor

PLOS ONE

Reviewers' comments:

Reviewer's Responses to Questions

**Comments to the Author**

1. If the authors have adequately addressed your comments raised in a previous round of review and you feel that this manuscript is now acceptable for publication, you may indicate that here to bypass the “Comments to the Author” section, enter your conflict of interest statement in the “Confidential to Editor” section, and submit your "Accept" recommendation.

Reviewer #1: (No Response)

2. Is the manuscript technically sound, and do the data support the conclusions?

Reviewer #1: Yes

3. Has the statistical analysis been performed appropriately and rigorously? 

Reviewer #1: Yes

4. Have the authors made all data underlying the findings in their manuscript fully available?

Reviewer #1: Yes

5. Is the manuscript presented in an intelligible fashion and written in standard English?

Reviewer #1: No

6. Review Comments to the Author

Reviewer #1: I read with interest this new version of the manuscript “Comparisons of fatigue between dialysis modalities: A cross-sectional study” by Maruyama and colleagues.

The questions and concerns raised in the first review process have been answered and addressed. However, I have a few additional comments about this new version.

1) The main aim of this study was to evaluate fatigue across different dialysis modalities. When adjusted for patient characteristics, no association was found between occurrence and severity of fatigue and dialysis modality. The authors should review the conclusions section in the abstract and the main article as the current wording is hard to understand and does not clearly describe the findings.

2) In the second paragraph of the Introduction, the authors refer to “patients on” HD and PD and the different hypotheses for the mechanisms of fatigue of each modality. The causes described seem to refer to the modality themselves, and not the patients.

3) At the end of the Introduction section, the aim is described as the comparison of prevalence OR severity. This should be modified to “and” as the authors evaluated both the prevalence and the severity with 2 different regression models.

4) The last sentence of the introduction is misleading as the authors state that the findings could “indicate the different pathophysiology of fatigue by dialysis modalities”. Although the findings might raise new questions and hypotheses on the pathophysiology of fatigue across modalities (with the differences seen between on and off dialysis days in HD and HDF), this study is not designed to evaluate the pathophysiology of fatigue.

5) The authors indicate that their original fatigue scale has been previously used in other studies, but they do not indicate if it has been validated as an accurate scale. The authors could specify in the Methods section why they have also used their original scale (easy to use? readily available?) if the aim of the study was not to validate this new scale.

6) The authors have indicated in this new version of the manuscript that their original scale was used on both on- and off-dialysis days in HD and HDF, with differences in the scores obtained. However, the authors do not mention if the scale was also used on different days in the combination therapy (HD+PD) group. Since the study aimed to compare the modalities and seemed to raise differences in fatigue depending on the timing of dialysis session, it would have been relevant to test whether fatigue also varied depending on timing in the combination therapy group, in which patients are treated alternatively with PD and HD. As timing of assessment of fatigue is not mentioned in this group, it brings some doubts about the reliability of the findings in this group if the timing related to HD sessions was not the same for all patients (for example: a patient assessed on HD day vs. a patient assessed after 3 PD days and due for HD 3 days later). If timing of assessment was different across the groups, this should be mentioned in the discussion.

In conclusion, this study reports interesting findings on fatigue in dialysis. The use of the original scale is interesting, but needs validation. The aim of the study was to compare fatigue across modalities. The findings related to the original scale are therefore hard to interpret as this scale has not been formally validated/compared to other scales.

The differences seen between on- and off-dialysis days also raise hypotheses on differences in the pathophysiology of fatigue across dialysis modalities. The inclusion of patients on combination therapy in this study is interesting (as this modality is mostly used in Japan). However, the methodology described did not permit to fully evaluate the differences across modalities without major confounding (including the timing of assessment in that group). Perhaps this would warrant further research in this group of patients who alternate between HD and PD.

7. PLOS authors have the option to publish the peer review history of their article (what does this mean?). If published, this will include your full peer review and any attached files.

Reviewer #1: No

---

## [Author Response · Author response to Decision Letter 1]

1 Dec 2020

Response to Reviewers

The authors have addressed previous comments. However, the manuscript needs significant editing to comply with standard English. There are too many typographical or grammatical errors to list here. The manuscript is not publishable in the current state and will require major editing and revisions. The original data set referred to under “supporting information” was not available to the editor or reviewer and will need to be accessible in order to comply with publication criteria. In addition to the reviewer’s comment, there are additional issues identified that require clarification:

RESPONSE: Thank you for revising our manuscript. The comments were very helpful. We have worked through each of the comments to address the issues raised, and a native English speaking editor has checked our revised manuscript. We have attached "S1 Table Original data set" as Supporting Information. We can release the data set used during the current study if requested. 

Abstract:

Please avoid vague statements in the abstract like “The scores of the POMS, VAS, and our original scale of fatigue were weakly but significantly inter-related”. The actual results would be more helpful. 

RESPONSE: As suggested, we have added the actual results as follows: "(rho =0.58, P<0.01; rho=0.47, P<0.01, and rho=0.42, P<0.01 between POMS and VAS, POMS and our original scale for fatigue, and VAS and our original scale for fatigue, respectively)."

“In the fatigue scales of on dialysis day by 3 tests, there were no statistical differences by modalities.” The meaning of this sentence is unclear. Please clarify and revise.

RESPONSE: Thank you for this helpful comment. In the Methods, we clarified that we measured our original scale of fatigue both on dialysis and dialysis-free day among patients on HD and online HDF. Additionally, we revised this sentence as follows: “The scores of these 3 tests showed no significant differences among the 4 modalities.”.

“The similar result was obtained in 122 patients without depression.” “The” should be removed.

RESPONSE: As suggested, we removed “The” from this sentence.

The conclusion of the abstract and manuscript is unclear and needs rephrasing. What do the authors mean with “The results indicated the least impact of dialysis modalities on fatigue in dialysis patients including HD, online HDF, PD and combined therapy with PD and HD.”?

RESPONSE: Thank you for this helpful comment. As suggested, we clarified and revised this sentence as follows: “The results showed that the types of dialysis modality had no significant effect on fatigue”.

Methods: “All participants read and wrote questionnaires for themselves”. Participants presumably did not write the questionnaires but responded to the questions without assistance.

RESPONSE: As suggested, we revised the word from “wrote” to “responded”.

The references need reformatting.

RESPONSE: As suggested, we have reformat the references using Endnote.

6. Review Comments to the Author

Reviewer #1: I read with interest this new version of the manuscript “Comparisons of fatigue between dialysis modalities: A cross-sectional study” by Maruyama and colleagues.

The questions and concerns raised in the first review process have been answered and addressed. However, I have a few additional comments about this new version.

RESPONSE: Thank you for revising our manuscript. The comments were very helpful.

1) The main aim of this study was to evaluate fatigue across different dialysis modalities. When adjusted for patient characteristics, no association was found between occurrence and severity of fatigue and dialysis modality. The authors should review the conclusions section in the abstract and the main article as the current wording is hard to understand and does not clearly describe the findings.

RESPONSE: Thank you for this helpful comment. As editor had also suggested, we clarified and revised this sentence as follows: “The results showed that the types of dialysis modality had no significant effect on fatigue”.

2) In the second paragraph of the Introduction, the authors refer to “patients on” HD and PD and the different hypotheses for the mechanisms of fatigue of each modality. The causes described seem to refer to the modality themselves, and not the patients.

RESPONSE: We completely agree with this comment. We revised the word from “patient on HD (or PD) could induce fatigue” to “HD (or PD) per se could induce fatigue”.

3) At the end of the Introduction section, the aim is described as the comparison of prevalence OR severity. This should be modified to “and” as the authors evaluated both the prevalence and the severity with 2 different regression models.

RESPONSE: As suggested, we revised the word from “OR” to “AND”.

4) The last sentence of the introduction is misleading as the authors state that the findings could “indicate the different pathophysiology of fatigue by dialysis modalities”. Although the findings might raise new questions and hypotheses on the pathophysiology of fatigue across modalities (with the differences seen between on and off dialysis days in HD and HDF), this study is not designed to evaluate the pathophysiology of fatigue.

RESPONSE: We completely agree with this comment that we could not clarify the pathophysiology of fatigue using this study design. We have deleted this sentence.

5) The authors indicate that their original fatigue scale has been previously used in other studies, but they do not indicate if it has been validated as an accurate scale. The authors could specify in the Methods section why they have also used their original scale (easy to use? readily available?) if the aim of the study was not to validate this new scale.

RESPONSE: Thank you for raising this important point. Although our original scale of fatigue has been previously used in other studies, we could not validate the effectiveness of this scale using this study design, as we described in the Limitation. The reasons why we used this scale were usability and a lower cost than other scales. As suggested, we have added following sentences to the Materials and methods section: “Although the effectiveness of this scale has not yet been validated, we used this scale because of its usability and lower cost than other scales”.

6) The authors have indicated in this new version of the manuscript that their original scale was used on both on- and off-dialysis days in HD and HDF, with differences in the scores obtained. However, the authors do not mention if the scale was also used on different days in the combination therapy (HD+PD) group. Since the study aimed to compare the modalities and seemed to raise differences in fatigue depending on the timing of dialysis session, it would have been relevant to test whether fatigue also varied depending on timing in the combination therapy group, in which patients are treated alternatively with PD and HD. As timing of assessment of fatigue is not mentioned in this group, it brings some doubts about the reliability of the findings in this group if the timing related to HD sessions was not the same for all patients (for example: a patient assessed on HD day vs. a patient assessed after 3 PD days and due for HD 3 days later). If timing of assessment was different across the groups, this should be mentioned in the discussion.

RESPONSE: Thank you for pointing out a serious flaw. As we described in the Materials and methods section, patients receiving combined therapy with PD and HD answered all self-reported questionnaires not at day of HD session but at regular outpatient visits. Therefore, as the reviewer concerned, impact of HD on the score of questionnaires was unclear, because timing of assessment related to HD sessions differed among patients in this group. We have added following sentences to the Limitation: “Since patients receiving combined therapy with PD and HD answered all self-reported questionnaires not on the HD session days but at regular outpatient visits, the impact of HD on the questionnaire scores was unclear, because the timing of assessment related to HD sessions differed among patients in this group.”

In conclusion, this study reports interesting findings on fatigue in dialysis. The use of the original scale is interesting, but needs validation. The aim of the study was to compare fatigue across modalities. The findings related to the original scale are therefore hard to interpret as this scale has not been formally validated/compared to other scales.

RESPONSE: We completely agree with this comment. Most serious limitation of this study was the lack of validity of our original scale. However, association between the prevalence and severity of fatigue and dialysis modalities were similar among three different scale of fatigue, including our original scale. Further studies will be needed to clarify not only the difference of fatigue by dialysis modalities, but also validity of the effectiveness of our original scale.

The differences seen between on- and off-dialysis days also raise hypotheses on differences in the pathophysiology of fatigue across dialysis modalities. The inclusion of patients on combination therapy in this study is interesting (as this modality is mostly used in Japan). However, the methodology described did not permit to fully evaluate the differences across modalities without major confounding (including the timing of assessment in that group). Perhaps this would warrant further research in this group of patients who alternate between HD and PD.

RESPONSE: Thank you for this helpful comment. As we described before, patients receiving combined therapy with PD and HD answered all self-reported questionnaires not at day of HD session but at regular outpatient visits. Therefore, impact of HD on the score of questionnaires was unclear, because timing of assessment related to HD sessions differed among patients in this group. We have added this information to the Limitation. As the reviewer pointed, further research investigating the effects of PD and those of HD on fatigue among patients receiving combined therapy is warranted.

---

## [Decision Letter · Decision Letter 2]

13 Jan 2021

PONE-D-20-17573R2

Comparisons of fatigue between dialysis modalities: A cross-sectional study

PLOS ONE

Dear Dr. Maruyama,

Thank you for submitting your manuscript to PLOS ONE. After careful consideration, we feel that it has merit but does not fully meet PLOS ONE’s publication criteria as it currently stands. Therefore, we invite you to submit a revised version of the manuscript that addresses the points raised during the review process.

The suggested minor revisions are included in the edited tracked version attached and relate predominantly to the conclusion of the abstract and the manuscript. Further minor edits and grammatical corrections have been made throughout the manuscripts.

We look forward to receiving your revised manuscript.

Kind regards,

Andrea K. Viecelli

Academic Editor

PLOS ONE

Reviewers' comments:

Reviewer's Responses to Questions

**Comments to the Author**

1. If the authors have adequately addressed your comments raised in a previous round of review and you feel that this manuscript is now acceptable for publication, you may indicate that here to bypass the “Comments to the Author” section, enter your conflict of interest statement in the “Confidential to Editor” section, and submit your "Accept" recommendation.

Reviewer #1: All comments have been addressed

2. Is the manuscript technically sound, and do the data support the conclusions?

Reviewer #1: Yes

3. Has the statistical analysis been performed appropriately and rigorously? 

Reviewer #1: Yes

4. Have the authors made all data underlying the findings in their manuscript fully available?

Reviewer #1: Yes

5. Is the manuscript presented in an intelligible fashion and written in standard English?

Reviewer #1: Yes

6. Review Comments to the Author

Reviewer #1: (No Response)

7. PLOS authors have the option to publish the peer review history of their article (what does this mean?). If published, this will include your full peer review and any attached files.

Reviewer #1: No

---

## [Author Response · Author response to Decision Letter 2]

14 Jan 2021

The suggested minor revisions are included in the edited tracked version attached and relate predominantly to the conclusion of the abstract and the manuscript. Further minor edits and grammatical corrections have been made throughout the manuscripts.

RESPONSE: Thank you for revising our manuscript. As suggested, we demonstrated data on age, gender and the proportion of diabetes only in the Result section. We revised the word from “native dialysis modality” to “preferred dialysis modality” in the third paragraph of the Discussion section. Additionally, we agree with the rephrased sentence in the last of this paragraph. We reformatted the list of references.

---

## [Editor Report · Decision Letter 3]

21 Jan 2021

PONE-D-20-17573R3

Comparisons of fatigue between dialysis modalities: A cross-sectional study

PLOS ONE

Dear Dr. Maruyama,

Thank you for submitting your manuscript to PLOS ONE. After careful consideration, we feel that it has merit but does not fully meet PLOS ONE’s publication criteria as it currently stands. Therefore, we invite you to submit a revised version of the manuscript that addresses the points raised during the review process.

Suggested revisions have been attached (Edited revision 3) to ensure the conclusions are supported by the data (please note the current version implies a causal relationship rather than an association). Comments have been added in the manuscript version attached to outline further minor revisions including the limitation of the small sample size and improvement of the first paragraph in the discussion to reflect a concise summary of the findings. Without these amendments, the manuscript does not fulfil the publication standards. The editor encourages the authors to address the suggested edits and revisions.

We look forward to receiving your revised manuscript.

Kind regards,

Andrea K. Viecelli

Academic Editor

PLOS ONE

---

## [Author Response · Author response to Decision Letter 3]

26 Jan 2021

Response to Reviewers

Suggested revisions have been attached (Edited revision 3) to ensure the conclusions are supported by the data (please note the current version implies a causal relationship rather than an association). Comments have been added in the manuscript version attached to outline further minor revisions including the limitation of the small sample size and improvement of the first paragraph in the discussion to reflect a concise summary of the findings. Without these amendments, the manuscript does not fulfil the publication standards. The editor encourages the authors to address the suggested edits and revisions.

RESPONSE: Thank you for revising our manuscript. As suggested, we amended our manuscript. First, we modified conclusion both of abstract and text to avoid any mention of causal relationship. Second, we modified the first paragraph of the Discussion. Third, we added limitation regarding the small sample size. Additionally, we approved all modifications.

---

## [Editor Report · Decision Letter 4]

28 Jan 2021

Comparisons of fatigue between dialysis modalities: A cross-sectional study

PONE-D-20-17573R4

Dear Dr. Maruyama,

We’re pleased to inform you that your manuscript has been judged scientifically suitable for publication and will be formally accepted for publication once it meets all outstanding technical requirements.

Kind regards,

Andrea K. Viecelli

Academic Editor

PLOS ONE
---

## [Editor Report · Acceptance letter]

1 Feb 2021

PONE-D-20-17573R4 

Comparisons of fatigue between dialysis modalities: A cross-sectional study 

Dear Dr. Maruyama:

I'm pleased to inform you that your manuscript has been deemed suitable for publication in PLOS ONE. Congratulations! Your manuscript is now with our production department. 

Kind regards, 

on behalf of

Dr. Andrea K. Viecelli 

Academic Editor

PLOS ONE